# Non-lytic clearance of influenza B virus from infected cells preserves epithelial barrier function

Rebekah E. Dumm[1], Jessica K. Fiege[2], Barbara M. Waring[2], Chay T. Kuo[3,4], Ryan A. Langlois[2] & Nicholas S. Heaton[1]

Influenza B virus (IBV) is an acute, respiratory RNA virus that has been assumed to induce the eventual death of all infected cells. We and others have shown however, that infection with apparently cytopathic viruses does not necessarily lead to cell death; some cells can intrinsically clear the virus and persist in the host long-term. To determine if any cells can survive direct IBV infection, we here generate a recombinant IBV capable of activating a host-cell reporter to permanently label all infected cells. Using this system, we demonstrate that IBV infection leads to the formation of a survivor cell population in the proximal airways that are ciliated-like, but transcriptionally and phenotypically distinct from both actively infected and bystander ciliated cells. We also show that survivor cells are critical to maintain respiratory barrier function. These results highlight a host response pathway that preserves the epithelium to limit the severity of IBV disease.

[1] Department of Molecular Genetics and Microbiology, Duke University School of Medicine, Durham, NC 27710, USA. [2] Department of Microbiology and Immunology, Center for Immunology University of Minnesota, Minneapolis, MN 55455, USA. [3] Department of Cell Biology, Duke University School of Medicine, Durham, NC 27710, USA. [4] Department of Neurobiology, Duke University School of Medicine, Durham, NC 27710, USA. Correspondence and requests for materials should be addressed to N.S.H. (email: nicholas.heaton@duke.edu)

Influenza viruses cause acute respiratory disease in up to 20% of the global population annually[1]. Influenza A virus (IAV) and influenza B virus (IBV) are the two genera of this family that cause the majority of disease in humans. Despite causing up to 45% of annual influenza-induced mortality[2], IBV has been relatively understudied compared to IAV. Although highly related, IAV and IBV are molecularly distinct in their protein products[3,4], tropisms[5,6], and have been shown to induce different antiviral responses[7,8]. Clinically, it has traditionally been assumed that IBV induces a milder form of disease. However, several recent epidemiological studies suggest that IBV disease can be just as severe as that induced by IAV in terms of clinical symptoms and outcomes[9–12]. Thus, a more complete understanding of the mechanisms of IBV disease is highly relevant.

In the lung, influenza viruses cause widespread cell death and changes to the structure and composition of the epithelium[13,14]. This tissue damage, combined with the rapid influx of immune cells and inflammatory cytokines, underlies the clinical symptoms of influenza disease. Because the lung epithelium is the first line of defense against incoming debris and pathogens, an ineffective epithelial barrier leaves the host susceptible to respiratory deficits, decreased mucociliary clearance and secondary infections. Previously, it has been thought that virus and immune-induced cell death account for all of the epithelial disruption observed during and after influenza virus infection. There is emerging evidence, however, that the mechanisms of epithelial barrier maintenance during infection may be more nuanced than previously appreciated.

While acute viral infections have been thought to uniformly lead to the lysis of infected cells, we and others have demonstrated that cells can non-lytically clear viral replication and survive direct infection with orthomyxo-, corona-, and rhabdoviruses[15–18]. Interestingly, these "survivor" cells appear to persist in the host long-term; however, for the most part, their effects on host physiology are unclear[19]. A number of reports have shown striking changes to respiratory epithelium during and after influenza virus infection;[13,14] in particular, a significant reduction in the number of ciliated cells has been reported[20]. However, there has not been a previous examination of whether cellular survival occurs after direct IBV infection. The mechanisms for how respiratory barrier function is maintained in the face of significant cellular damage are incompletely understood, and the potential contributions of cells that can survive direct infection have not been evaluated.

In this report, we first test if cellular survival can occur after IBV infection. To accomplish this, we generate a Cre-expressing reporter virus in the B/Malaysia/2506/2004 background. We use this tool to demonstrate that epithelial cells are capable of surviving IBV infection in mice. We report that the majority of the cells that survive IBV infection are ciliated-like cells that display significant transcriptional alterations relative to bystander ciliated cells. These transcriptional changes correlate with a number of unique cellular morphology changes such as the absence of apical cilia. Upon depletion of the survivor cell population, we observe increased epithelial permeability, decreased pulmonary compliance, and delayed recovery from infection. Based on these data, we propose a model in which non-lytic clearance of IBV and subsequent cellular survival is a host-adaptive process to preserve critical respiratory barrier function during an acute viral infection.

## Results

### Generation of a Cre-expressing influenza B virus.
In order to determine if any cells could survive direct IBV infection, we generated a Cre recombinase reporter virus in the B/Malaysia/

2506/2004 (Mal/04) background. We accomplished this by encoding Cre recombinase in the polymerase (PB1) segment of the viral genome (Fig. 1a), an approach that we have previously published to be appropriate for exogenous gene expression in IBV[21]. When this reporter virus infects a cell with a Cre-responsive cassette, it removes the STOP cassette flanked by loxP sites to allow constitutive expression of the reporter protein and permanently labels any cell that has been infected (Fig. 1b). After growth in embryonated chicken eggs, the resulting titers were similar to wild-type Mal/04 (Fig. 1c). To confirm functional expression of Cre recombinase during viral infection, we infected transgenic A549s, a lung epithelial cell line with an integrated lox-STOP-lox-zsGreen cassette. These cells express the fluorophore zsGreen after a Cre-mediated recombination event[22]. After infection with the wild-type (WT) Mal/04 or Mal/04-Cre reporter virus, expression of zsGreen was observed exclusively with the Cre-expressing virus (Fig. 1d).

To determine the stability of the Cre insert in the viral genome, we performed serial passaging of the virus in eggs and found that both the insert length and Cre recombinase activity were maintained over four passages (Fig. 1e, f). When we examined infected cells for Mal/04 protein and zsGreen expression at 72 hours post-infection, we found that ~85% were co-positive; rare cells with viral protein that were zsGreen negative are likely reflective of particularly high levels of viral replication and more complete host shutoff (Supplementary Fig. 1). We next assessed viral growth kinetics in a multicycle growth curve. The WT and Cre-expressing viruses grew at similar rates, however, we observed a slight reduction in the peak titer of the Mal/04-Cre virus (Fig. 1g). Finally, to determine the virulence of the Cre-expressing virus, we infected mice with a range of doses of both Mal/04-Cre and wild-type virus. The Cre-expressing virus induced morbidity in the animals (Fig. 1h) albeit with a higher median lethal dose relative to WT Mal/04. This level of attenuation, however, is similar to previously described influenza reporter viruses[23,24].

### A population of cells in the upper airways survive IBV infection.
To first assess the cellular tropism of Mal/04 in vivo, we generated a green fluorescent Mal/04-mNeon to label actively infected cells. This virus encodes the mNeon gene in segment 4, a genetic engineering strategy we have previously described[3]. We then infected mice and analyzed mNeon positive cells for EpCam and CD45, cell surface markers specific for epithelial and hematopoietic lineages, respectively. We collected cells from infected mice at 2 days post-infection (DPI) and found that ~47% of mNeon + cells were epithelial while ~23% were hematopoietic in origin (Fig. 2a, b and Supplementary Fig. 2A, B). While EpCam + mNeon + cells are almost certainly directly infected, this assay does not distinguish between directly infected CD45 + cells or those that appear positive after phagocytosing mNeon positive cells[23]. mNeon + cells that are both CD45- EpCam- are most likely a combination of mesenchymal cells such as fibroblasts and cells that lost their surface markers due to the enzymatic digestion protocol.

To test for cellular survival after Mal/04-Cre infection in vivo, we infected transgenic mice containing a lox-STOP-lox-tdTomato cassette to fluorescently label cells that have been infected with the Cre-expressing virus. To define survivor cell populations, we collected lungs from mice at 14 days post-infection and looked for tdTomato positive cells. We chose this timepoint after experimentally determining that Mal/04-Cre virus is eliminated from the host by 9 days post-infection (Supplementary Fig. 2C). Using flow cytometry, we found that a number of cells, in fact, do survive IBV infection (Fig. 2c). Further, cells non-uniformly

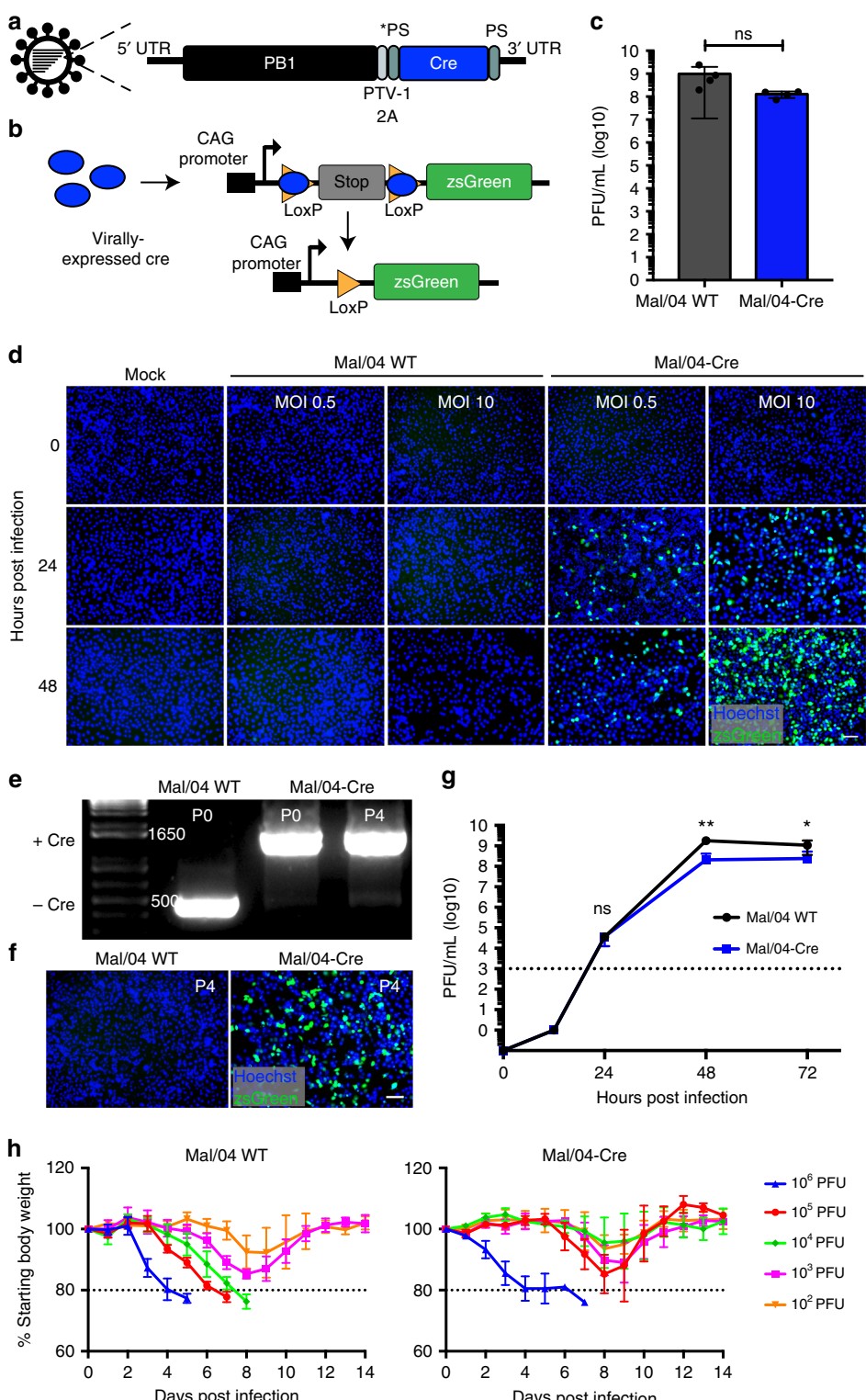

survive viral infection, as the composition of the surviving population was radically different that the total infected cell population, with > 90% of surviving cells being epithelial in origin (Fig. 2c, d). In order to quantify the percent of the epithelium that had been infected and survived, we gated specifically on EpCam and observed that ~3% of epithelial cells express tdTomato after Mal/04-Cre infection (Fig. 2e, f). To understand the localization of survivor cells in the lungs, we next performed lung sectioning from the trachea, bronchioles and alveoli and observed a high

percentage of survivor cells in the trachea (Fig. 2g). This result is different from our previous work with IAV, which had identified club cells enriched in the small distal airways as the major surviving cell type[22].

To better characterize the spatial relationship between IBV survivor cells, we collected tracheas from infected mice, opened and flattened the tissue, and imaged the epithelium from the apical side as a whole mount. This analysis revealed that while survivor cells were stochastically distributed across the trachea,

**Fig. 1** Design and characterization of a Cre-expressing influenza B virus. **a** Schematic of B/Malaysia/2506/2004 engineered virus containing Cre recombinase. PB1 polymerase segment 1, UTR untranslated region, PS packaging signal, PTV-1 2A porcine teschovirus 2A sequence. **b** Schematic of Cre recombination of the reporter construct allowing expression of zsGreen. **c** Quantification of viral titer after growth in embryonated chicken eggs of the Mal/04-Cre compared to wild-type virus. PFU plaque forming units. $n = 4$ replicates per sample, Student's T-test **d** Microscopy time course of viral replication of Mal/04 WT and Mal/04-Cre in vitro using the human lung epithelial cell line, A549, harboring a floxed zsGreen fluorescent marker. MOI multiplicity of infection. Scale bars = 10 μm. **e** Agarose gel demonstrating maintenance of Cre recombinase insert in Mal/04-Cre virus across four serial passages (P4) in embryonated chicken eggs. **f** Microscopy of A549 epithelial cell expression of the zsGreen Cre reporter after infection with the passage four (P4) viruses. Scale bars = 10 μm. **g** Time course of viral replication in embryonated chicken eggs, quantified by plaque assay at the indicated timepoints. $n = 3$ eggs per timepoint for each sample, two-way ANOVA. Dashed line = limit of detection for virus using plaque assays. **h** Body weight of WT C57BL/6 infected with Mal/04 WT and Mal/04-Cre and monitored for 2 weeks post-infection for weight loss. $n = 5$ mice per timepoint. Dashed line = 80% body weight as humane endpoint. S.E.M. was plotted to indicate variability around the mean for tested samples, $*p \leq 0.05$, $**p \leq 0.001$, ns = not significant

their total surface area accounted for 3.32% of the apical epithelium (Fig. 3a, b). To determine the cellular composition of the survivor cell population in the proximal airways, we collected survivor cells and performed flow cytometry analysis. We utilized the surface-expressed markers (GSIB4, SSEA-1, and CD24) that have been shown to correlate with the major cell types (basal, secretory, and ciliated, respectively) in the trachea[25–27]. We found populations of tdTomato + survivor cells labeled with the secretory marker SSEA-1 and the basal cell marker GSIB4 (Fig. 3c, d). Survivor cells expressing the ciliated cell surface marker CD24, however, were the largest component of the population (Fig. 3c, d). To support our flow cytometry analysis, we collected mice 14 days post-infection with Mal/04-Cre and cross-sectioned tracheas to stain for the epithelial markers SSEA-1, KRT5 and FOXJ1 using microscopy. We counted tdTomato + cells across independent experiments and found that while few survivor cells expressed SSEA-1 (secretory cells) or KRT5 (basal stem cells), 74% of tdTomato + survivor cells expressed FOXJ1, a well characterized marker for ciliated cells (Fig. 3e–g). Since these FOXJ1 + cells comprise the vast majority of the survivor cell population, we decided to focus our analysis on this cell type.

**Survivor cells acquire a unique gene expression profile**. We next wanted to determine if surviving viral infection induced any physiological alterations to FOXJ1 + survivor cells. Ciliated cells are terminally differentiated with a well-defined gene expression profile[28–30], however, influenza viruses induce host-transcriptional shutoff and have the potential to downregulate cell-specific markers during infection[31–33]. Previous studies have utilized CD24(high) as a surface-expressed marker for flow cytometry identification of ciliated cells[25–27,34–39], suggesting that this marker may be useful for further study of surviving FOXJ1 + cells. We first verified that CD24 did indeed specifically label ciliated cells by co-staining tracheal tissue with canonical cell markers (Supplementary Fig. 3). Next, to determine if viral infection would suppress the expression of CD24 on these cells, we performed single-cell RNA-sequencing of infected and uninfected murine epithelial lung cells. Unbiased clustering of cell types based on their gene expression profiles verified that CD24 expression is not downregulated by influenza virus infection (Supplementary Fig. 4A–F). Additionally, via immunostaining of tracheal tissue, we verified that survivor FOXJ1 + cells also express CD24 at 14 DPI (Supplementary Fig 4G, H).

Following our validation of CD24 as a ciliated cell marker, we next performed RNA sequencing to define any virally induced transcriptional alterations in survivor cells. We sorted CD24 + cells from the lungs of animals: prior to infection (0 DPI), during active infection (2 DPI), and matched populations at 14 days post-infection that had either experienced direct viral infection (14 DPI tdTomato + ) or those that had never been infected (14 DPI tdTomato−) (Fig. 4a). In order to verify

infection, we first confirmed that viral RNA was present in the samples at 2 days post-infection then cleared by 14 days post-infection in the survivor cell samples (Fig. 4b). Differential gene expression analysis data (Fig. 4c and Supplementary Table 1) revealed highly divergent gene expression profiles across the different ciliated cell groups. We quantified the relationship between samples and variable gene expression using the Spearman's rank correlation coefficient (Fig. 4d). As expected, the uninfected ciliated cells from both the mock infected animals and the animals at 14 DPI were highly similar but distinct from the actively infected cells (2 DPI). However, the survivor cells were highly divergent from both the uninfected ciliated cells as well as the actively infected ciliated cells and acquire a unique transcriptional profile.

The transcriptional changes observed in survivor cells could theoretically be the result of continued expression of antiviral or stress response genes. Analysis of the top 100 genes upregulated at 2 DPI, however, revealed that these genes returned to normal levels in survivor cells in parallel with the increase and loss of Mal/04 viral RNA (Fig. 4e and Supplementary Table 2). We performed an unbiased gene ontology analysis using DAVID software[40,41] to classify pathways dysregulated between uninfected and survivor cells at 14DPI based on the top 500 significant differentially expressed genes. Some pathways, which are not typically highly active in ciliated cells, such as surfactant homeostasis[42] were upregulated uniquely in the survivor cell population (Supplementary Data 1 and Fig. 4f). This analysis also revealed that controllers of ciliated cell morphology and cilia structure/function-associated pathways were significantly down-regulated in survivor cells compared to uninfected cells (Supplementary Data 1 and Fig. 4f). Specifically, we found that *Foxj1*, *Rfx3*, and *Myb*, transcription factors known to be critical for the ciliated cell phenotype[43–45] and responsible for ciliogenesis and maintenance of cilia structure, were uniquely down-regulated in the survivor cell population (Fig. 4g).

To validate the RNA-sequencing data, we sorted murine ciliated cells based on CD24 and tdTomato as a marker of infection and performed quantitative reverse transcription PCR (qRT-PCR) for some of the genes most dysregulated in survivor cells. For the upregulated genes, we tested expression of the surfactant components *SftpA* and *SftpD*, as well as *Cldn18* and *Igta7*, which are both involved in epithelial cell-cell interactions. As expected, the transcription of these genes was upregulated in survivor CD24 + cells compared to uninfected CD24 + ciliated cells (Supplementary Fig. 5). As a representative gene, which was downregulated in survivor cells, we measured the transcript levels of *Foxj1*, the transcription factor responsible for ciliogenesis and cilia motility. We observed strong downregulation of *Foxj1* RNA specifically in survivor, but not uninfected, ciliated cells (Fig. 5a). To ensure that the decreased transcript levels was reflective of differences at the protein level, we performed antibody staining for FOXJ1

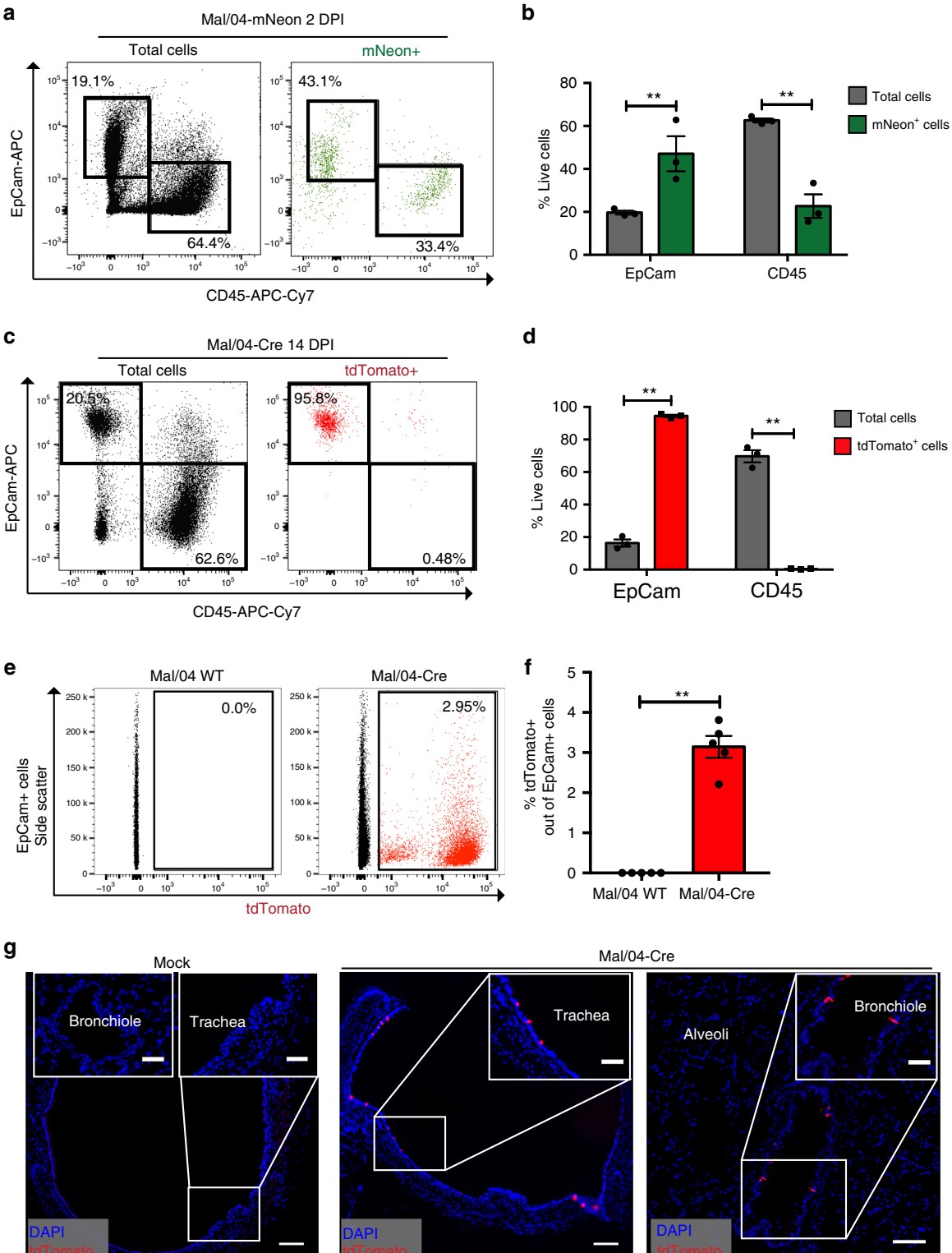

**Fig. 2** Epithelial cells are infected by IBV and survive in the large airways of mice. **a** Flow cytometry of mNeon− (total cells) and mNeon+ (infected cells) from the lungs of Mal/04-mNeon infected mice at 2 DPI, stained for EpCam (epithelial lineage) and CD45 (hematopoietic lineage). **b** Quantification of total and mNeon+ infected cell expression of EpCam and CD45. $n = 3$ mice per sample, two-way ANOVA. **c** Flow cytometry of tdTomato− (total cells) and tdTomato+ (survivor cells) from the lungs of Mal/04-Cre infected tdTomato Cre reporter transgenic mice at 14 DPI, stained for EpCam (epithelial lineage) and CD45 (hematopoietic lineage). **d** Quantification of total and tdTomato+ survivor cell expression of EpCam and CD45. $n = 3$ mice per sample, two-way ANOVA. **e** Flow cytometry of epithelial (EpCam+) tdTomato+ survivor cells from Mal/04 WT and Mal/04-Cre infected mouse lungs. **f** Quantification of tdTomato+ cells from murine lungs. $n = 5$ mice per sample, Student's $T$-test. **g** Microscopy of 14 DPI Mock and Mal/04-Cre infected murine lungs with subpanels showing airways (bronchiole and trachea) and alveolar region. Scale bar = 100 μm, inset scale bar = 50 μm. S.E.M. was plotted to indicate variability around the mean for tested samples, $*p \leq 0.05$, $**p \leq 0.001$, ns = not significant

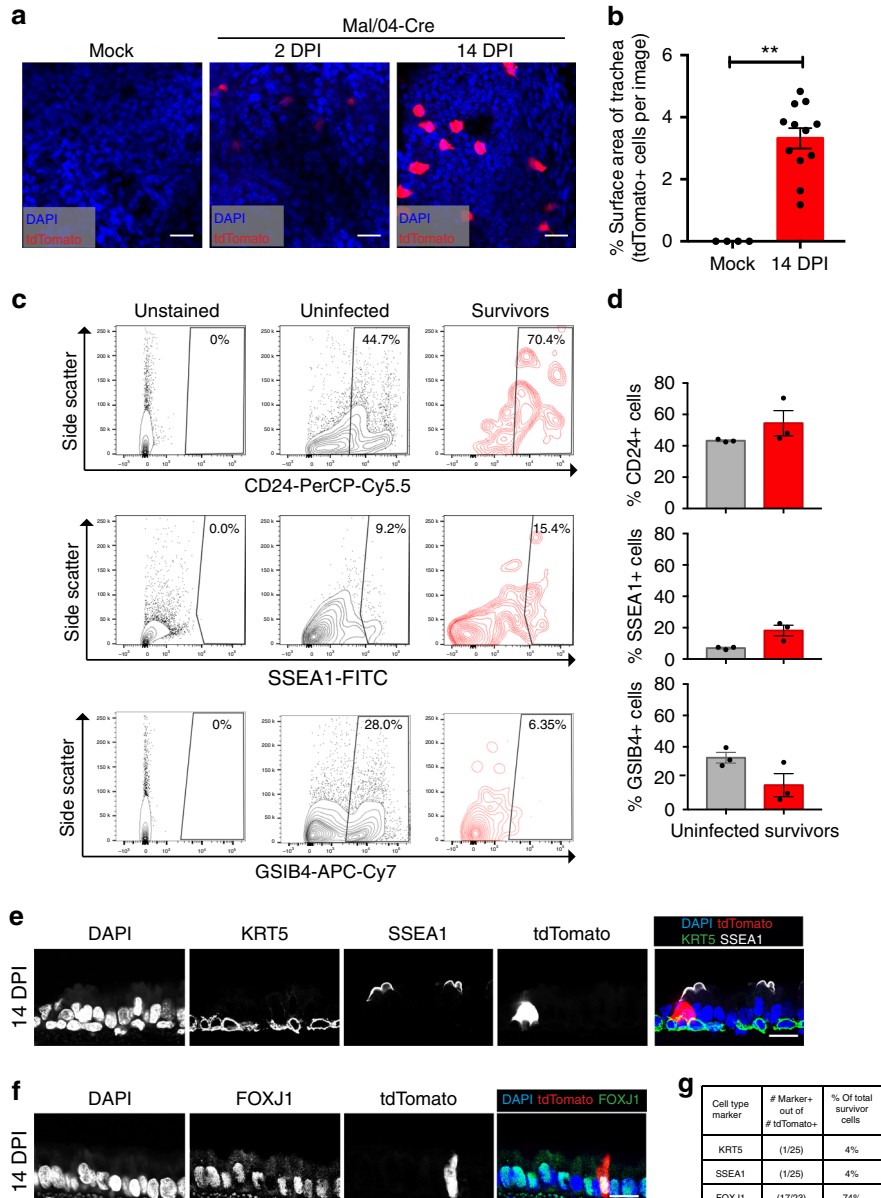

**Fig. 3** IBV survivor cell populations are predominantly composed of cells expressing ciliated cell markers. **a** Tracheas were collected from Mock and Mal/04-Cre infected mice at 2 and 14 DPI then opened longitudinally and flattened to image the surface area of the whole-mount tracheal epithelium. Scale bar = 25 μm. **b** Quantification of apical surface area covered by tdTomato+ survivor cells from whole-mount trachea images. $n = 4$, 12 images per group, respectively, Student's $T$-test. **c** Flow cytometry of 14 DPI tdTomato- (uninfected cells) and 14 DPI tdTomato+ (survivor cells) from the lungs of Mal/04-Cre infected mice, where cells are stained with CD24 (ciliated), SSEA-1 (secretory), GSIB4 (basal) cell surface markers. **d** Quantification of flow cytometry. $n = 3$ mice per group. **e** Cross-sectioned microscopy of murine tracheal epithelial cells from mice 14 days post-infection with Mal/04-Cre stained with KRT5 to label basal stem cells and SSEA-1 to label secretory cells. **f** Cross-sectioned microscopy of murine tracheal epithelial cells from mice 14 days post-infection with Mal/04-Cre stained with FOXJ1 to label ciliated cells. **g** Quantification of tdTomato+ survivor cells stained with cell-specific markers: KRT5, SSEA-1, and FOXJ1 compiled from two independent experiments. Data are presented as the number of marker+ cells out of total tdTomato+ cells and the percentage was calculated for marker+ cells out of total tdTomato+ cells imaged. S.E.M was plotted to indicate variability around the mean for tested samples, $*p \leq 0.05$, $**p \leq 0.001$, ns not significant

protein levels in cells infected with Mal/04 and again observed a specific decrease in signal (Fig. 5b, c).

**Survivor cells are morphologically distinct from bystander ciliated cells.** We next wanted to define if the altered gene expression profiles in survivor cells translated into phenotypic differences in this cell population relative to uninfected ciliated cells. Since genes related to cilia structure and function were

highly downregulated in survivor cells, we began by determining the presence of apical cilia on survivor and bystander ciliated cells in murine tracheal sections. Using acetylated tubulin as a marker for motile cilia, we found that cilia were readily detected on bona fide FOXJ1 + ciliated cells from uninfected mice, where ~91% of ciliated cells displayed mature surface cilia (Fig. 5d–f). In contrast, we found that only ~42% of survivor cells that were positive for FOXJ1 displayed mature apical cilia (Fig. 5g–i). We verified that cilia loss was not a generalized phenomenon in infected lungs, as

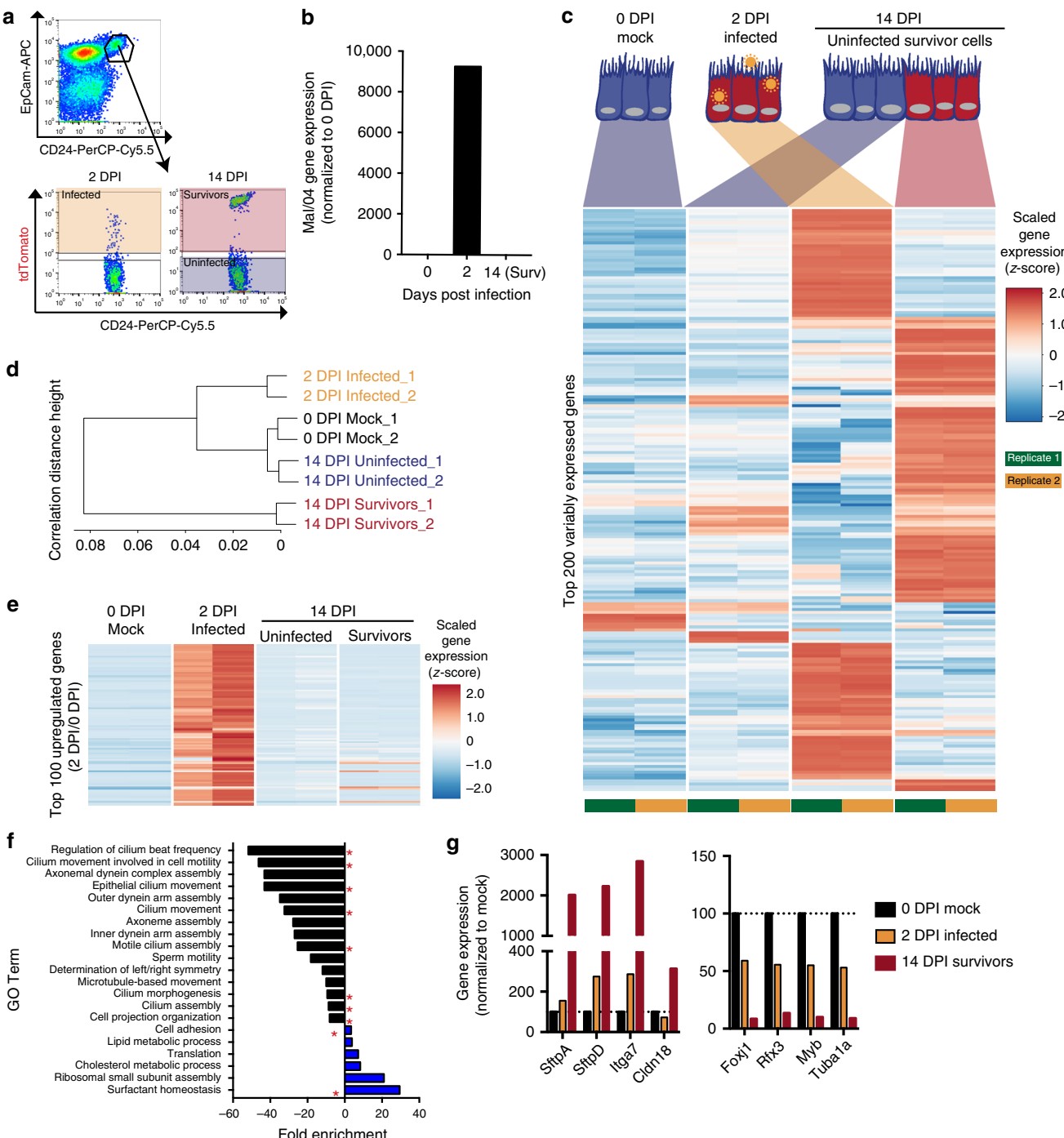

**Fig. 4** Actively infected and uninfected ciliated cells are transcriptionally distinct from matched populations of survivor cells. **a** FACS gating strategy to isolate CD45− EpCam+ CD24+ (ciliated) cells from the lungs of Mock and Mal/04-Cre infected mice. Cre-activated tdTomato expression was used to identify 2 DPI infected and 14 DPI survivor cells. **b** Viral mRNA gene expression of the Mal/04 genome in 2 DPI infected and 14 DPI survivor cells (Normalized to 0 DPI). **c** Heatmap of the top 200 variably expressed genes across the four conditions (eight samples total) from murine lung ciliated cells: 0 DPI mock, 2 DPI actively infected with virus, 14 DPI tdTomato− uninfected and 14 DPI tdTomato+ survivor cells (14 DPI samples collected from the same lungs). Gene expression has been normalized using DeSeq2's default scaling factor parameters and normalized using a shifted log transformation [$\log_2(n+1)$]. Expression was then scaled across rows and the relative $z$-scores plotted. **d** Measurement of correlation distance between the indicated samples using Spearman's rank test. **e** Heatmap of top 100 genes upregulated in 2 DPI cells compared to 0 DPI mock cells. Gene expression was then scaled across rows and the relative $z$-score plotted. **f** Gene Ontology of Biological Processes generated using DAVID software for genes differentially expressed between 14 DPI tdTomato− (uninfected cells) and 14 DPI tdTomato+ (survivor cells) red asterisks indicate GO processes that are associated with dysregulated cilia function or gene pathways not expected to be active in normal ciliated cells. **g** RNAseq gene expression, normalized to 0 DPI mock cells, for selected dysregulated genes derived from the pathways identified in the ontology analysis in **f**

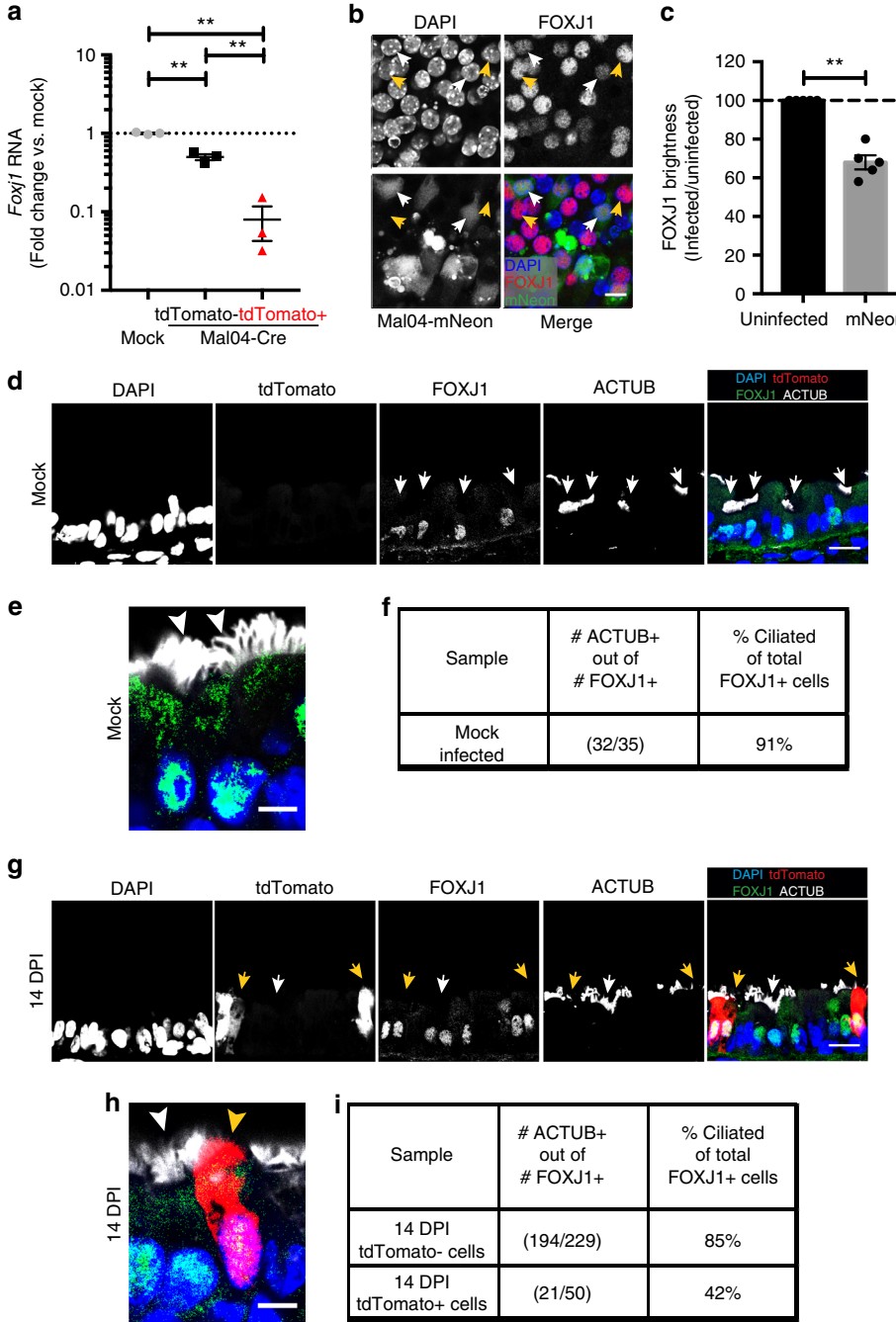

**Fig. 5** Ciliated-like survivor cells display reduced expression of ciliated cell transcription factors and fail to elaborate surface cilia. **a** Quantitative RT-PCR measuring RNA levels of *Forkhead Box J1 (Foxj1)* in flow sorted EpCam+ CD24+ mock infected cells, tdTomato− (uninfected cells), and tdTomato+ (survivor cells) at 14 days post-infection. $n = 3$ mice per sample, one-way ANOVA. **b** Air–liquid interface differentiated cultures of primary murine epithelial cells were infected with Mal/04-mNeon (green) and stained for FOXJ1 (red). White arrows = mNeon+ (infected cells), yellow arrows = mNeon− (uninfected cells). **c** Quantification of FOXJ1 brightness per cell, normalized to brightness of uninfected cells. $n = 6$ images, at least four cells quantified per image for each group, Student's *T*-test. **d** Microscopy of cross-sectioned murine tracheas from mock infected mice where epithelial cells were stained for FOXJ1 and acetylated tubulin for mature cilia. White arrow = tdTomato− (uninfected cells) with intact cilia. **e** Close-up panel of uninfected murine tracheal ciliated cells stained for FOXJ1 and ACTUB. White arrow = tdTomato− (uninfected cells) with intact cilia. **f** Quantification of uninfected cells that are double positive for FOXJ1 and ACTUB. **g** Microscopy of cross-sectioned murine tracheas from Mal/04-Cre infected mice at 14 DPI where epithelial cells were stained for FOXJ1 and ACTUB. White arrow = tdTomato− (uninfected cells) with intact cilia, yellow arrows = tdTomato+ (survivor cells) lacking surface cilia. **h** Close-up panel of murine tracheal uninfected Mal/04-Cre infected cells stained for FOXJ1 and ACTUB. White arrow = tdTomato− (uninfected cells) with intact cilia, yellow arrows = tdTomato+ (survivor cells) lacking surface cilia. **i** Quantification of Mal/04-Cre infected cells: tdTomato− (uninfected cells) and tdTomato+ (survivor cells) that are double-positive for FOXJ1 and ACTUB. S.E.M. was plotted to indicate variability around the mean for tested samples, *$p \leq 0.05$, **$p \leq 0.001$, ns not significant

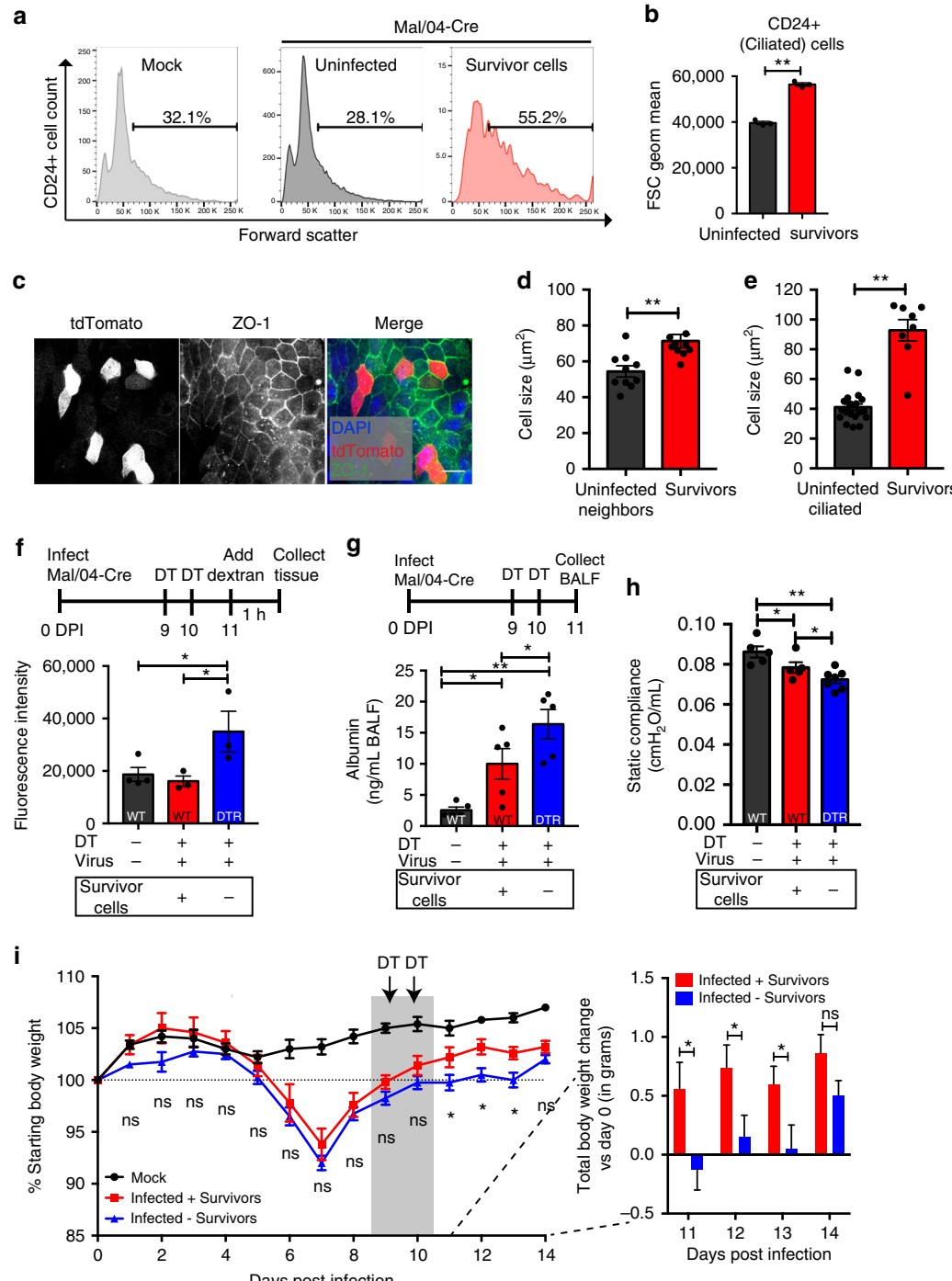

uninfected neighboring ciliated cells in Mal/04-Cre infected mice displayed mature surface cilia at similar frequencies (~85%) to ciliated cells from mock infected animals (Fig. 5g–i). The loss of surface cilia is consistent with previously reported epithelial alterations such as desquamation and denudation after influenza virus infections[13,14]. However, our data suggest that respiratory epithelium cilia loss is not solely the result of the lytic elimination of ciliated cells.

We were next interested in understanding how viral infection induced this population of phenotypically unique, FOXJ1 + survivor cells. We hypothesized that they could be derived from either the virus inducing cilia loss in mature ciliated cells or by preventing complete differentiation of progenitor cells into mature ciliated cells, or both. If these cells were derived from

incomplete differentiation, one would expect them to be the product of progenitor cell division[46] and also for the frequency of these ciliated-like survivor cells to increase over time. A time course analysis of survivor CD24 + cell frequency within the total survivor cell population revealed stability in the frequency of survivor ciliated-like cells (Supplementary Fig. 6A). Further, BrdU labeling analysis showed that this population of cells rarely divides and is not likely derived from proliferating progenitor cell populations (Supplementary Fig. 6B). Thus, we favor the model in which these morphological alterations are the result of viral effects on mature ciliated cells.

In addition to predicting changes to the expression and function of apical cilia, our transcriptional data also indicated that the cell-cell interaction proteins that control epithelial cell organization are

**Fig. 6** Survivor cells contribute to the maintenance of epithelial barrier integrity during IBV infection. **a** CD24+ (ciliated) cells were isolated from Mock and Mal/04-Cre infected murine lungs at 14 DPI and forward scatter (as a measure of cell volume) was measured for mock infected cells, tdTomato− (uninfected cells) and tdTomato+ (survivor cells). **b** Quantification of the geometric mean of cell forward scatter, $n = 3$ mice per group, Student's $T$-test. **c** Microscopy of flattened whole-mount tracheal epithelium at 9 DPI with survivor cells expressing tdTomato (red) stained with ZO-1 (green) for tight junctions. Scale bar = 10 μm. **d** Quantification of whole-mount microscopy cell size according to ZO-1 staining for tdTomato− (uninfected cells) and tdTomato+ (survivor cells) in the same image. $n = 10$ images, with at least four cells quantified per group in each image, Student's $T$-test. **e** Quantification of whole-mount microscopy cell size according to ZO-1 staining for ciliated tracheal cells from mock infected mice and tdTomato+ (survivor cells) from Mal/04-Cre infected mice. $n = 21$ uninfected ciliated cells, eight survivor cells from three images each. Student's $T$-test. **f–i** WT and DTR-transgenic mice were either mock or Mal/04-Cre infected and treated with diphtheria toxin (DT) to ablate survivor cells specifically in the DTR-transgenic mice). **f** Epithelial permeability was measured by administering 4 kDa fluorescein isothiocyanate (FITC)-Dextran and measuring leakage into the surrounding tissue at 11 DPI, 2 days after depletion of survivor cells. $n = 4, 3, 3$ mice per group, respectively, one-way ANOVA. **g** Albumin leakage into the bronchoalveolar lavage fluid (BALF) was measured by ELISA at 11 DPI, 2 days after depletion of survivor cells. $n = 5$ mice per group, one-way ANOVA. **h** Static compliance of lungs was measured using Flexivent physiology studies at 11 DPI, 2 days after depletion of survivor cells. $n = 5, 5, 7$ mice per group, respectively, two-way ANOVA. **i** Body weight curves of tdTomato and DTR-transgenic mice infected and treated with DT. $n = 5, 5, 4$ mice per group, respectively, one-way ANOVA at each timepoint. S.E.M. was plotted to indicate variability around the mean for tested samples, *$p \leq 0.05$, **$p \leq 0.001$, ns not significant

changed in survivor cells. We, therefore, infected tdTomato transgenic mice with Mal/04-Cre and collected CD24 + ciliated cells at 14 days post-infection. We then quantified the size of CD24 + survivor cells compared to matched uninfected cells from the same lungs and CD24 + cells from mock infected mice. Using forward scatter as a marker of cell size, we found that survivor cells were 43% larger than both cells from mock infected and uninfected cells from infected murine lungs (Fig. 6a, b). We then used a marker of tight junctions (ZO-1) to measure the apical cell size in flattened whole-mounted tracheas collected from Mal/04-Cre infected mice. Using this method, we observed a significant increase in the apical surface area of survivor cells relative to neighboring uninfected cells (Fig. 6c, d) and uninfected ciliated cells from an uninfected animal (Fig. 6e). In order to validate these results in a system where the kinetics of infection and survival are better controlled, we adapted a murine air–liquid interface (ALI) differentiated cell culture model to study cellular survival in vitro (Supplementary Fig. 7). We produced these cultures from uninfected Cre reporter transgenic mice so that, as with our in vivo system, survivor cells were irreversibly labeled with tdTomato after viral infection. We infected these cultures and at 14 days post-infection, after viral clearance, we observed similar changes to cellular morphology as with our in vivo studies (Supplementary Fig. 8), further supporting the dynamic ability of ciliated epithelial cells to respond to direct viral infection by altering cytoskeletal structures.

**Survivor cells contribute to the maintenance of epithelial barrier function**. Finally, we wanted to define a rationale for the maintenance of previously infected cells with altered transcriptional profiles in the respiratory epithelium. Our data suggested that cells were responding to viral infection by changing their shape and interactions with neighboring cells. Similar processes have been reported in gut epithelial cells, where inflammation, stress and pathogens cause disruption of the epithelial barrier[47,48]. In the lung, this regulation plays a role in the maintenance of barrier integrity and preservation of lung functions[48–51]. Clinically, the maintenance of the lung barrier is critical to preserve pulmonary functionality[48–51]. We, therefore, hypothesized that not only the presence, but also potentially the morphological changes in survivor cells could allow them to contribute to maintenance of the epithelial barrier during a highly destructive acute viral infection.

To determine if survivor cells contribute to epithelial barrier function in vivo, we first tested whether the presence of survivor cells helps prevent leakage from the large airway into the surrounding tissue. We, therefore, infected a Cre-responsive transgenic reporter mouse line wherein Cre activates expression

of the diphtheria toxin receptor (DTR). This line allows for the depletion of entire population of survivor cells via diphtheria toxin (DT) administration. For DT depletion, we chose a timepoint after the resolution of viral infection in vivo (Supplementary Fig. 2C), as earlier administration would also deplete all infected cells and likely alter the normal course of infection. As a control group, we infected and DT-treated non-transgenic mice that do not express the DTR on survivor cells after infection. After ablation of survivor cells, we made use of two complementary methods of assessing lung barrier function. First, we used a low molecular weight fluorescent dextran to assess solute and ion permeability[52]. To accomplish this, we administered a fluorescein isothiocyanate (FITC)-labeled dextran intranasally then collected the tracheas. After washing residual dextran from the lumen of the trachea, we analyzed the surrounding tracheal tissue for fluorescence. This analysis revealed that the leakage of particulate into tracheal tissue was significantly increased when survivor cells were absent from the large airways (Fig. 6f). We also verified the leakage phenotype in our ALI culture system (Supplementary Fig. 9). Next, as an organ-level measurement of barrier function, we analyzed the contribution of survivor cells to albumin (a plasma protein normally absent from healthy lungs) leakage into the lungs of infected animals. DTR-transgenic and WT animals were infected and treated with DT, and 24 hours post-depletion the bronchoalveolar lavage fluid (BALF) was collected. As expected when survivor cells were depleted, we observed significantly increased albumin leakage into the lung (Fig. 6g). These data indicate that survivor cells likely play a role in the maintenance of barrier function and, if they had not survived and been eliminated along with other infected cells, their loss would have decreased epithelial barrier integrity during active infection.

In addition to observing an epithelial barrier phenotype, we wanted to validate that the survivor cell contributions to the epithelium had a meaningful effect on lung function. We, therefore, utilized a forced oscillation technique to measure lung compliance in mice either harboring or depleted for survivor cells at 11 days post-infection. We were able to detect a small but reproducible decrease in overall lung compliance in the absence of survivor cells (Fig. 6h). These respiratory defects are consistent with injury models resulting in increased fluid leakage into the lung[53,54]. Further, the magnitude of the compliance phenotype is comparable to mouse models of asthma, fibrosis, and emphysema where 30% reductions in compliance were measured using the same Flexivent technique[55].

Finally, to show that the presence of survivor cells and their effects on barrier function were important for the recovery from influenza disease, we monitored the body weight of infected animals with and without survivor cells. After a sub-lethal

infection, we observed that the transgenic and control mice had equal morbidity (as measured by body weight loss) up to the point of viral clearance at 9 days post-infection. After toxin administration at days 9 and 10 post-infection, however, the recovery rates of the two groups diverged. The group with intact survivor cell populations continued to recover while mice without survivor cells had significantly delayed recovery from infection (Fig. 6i). DT treatment in the absence of DTR labeled cells had no effect on mouse body weight irrespective of the infection status of the mouse (Fig. 6i and Supplementary Fig. 10). These data together support a model in which the survival of cells in the large airways helps maintain barrier function and if lost, is detrimental to the host during influenza B virus infection.

## Discussion

Gas-exchange in the lung requires respiratory epithelial cells to maintain constant contact with the air and as a result, there is frequent contact with airborne pathogens. When the lung epithelium encounters a cytopathic pathogen such as influenza virus, a large number of epithelial cells can be damaged and eliminated. When this happens, pulmonary barrier function can be compromised, leading to fluid leaking into the airways and decreased gaseous exchange, potentially resulting in the host succumbing to the infection. To maintain lung function, there are a number of characteristic alterations to the epithelium that occur during and after viral infection. While epithelial lung damage has been studied extensively after influenza virus infection, models have traditionally assumed that changes to epithelium composition are solely the results of cellular death and regenerative potential of the epithelium.

Although there were no previous reports of any cell surviving IBV infection, we hypothesized that some epithelial cells would have the ability to survive direct viral infection, and that this population of survivor cells could play a role in viral pathogenesis. To test this hypothesis, we engineered an IBV strain to express Cre recombinase and found that epithelial cells can survive IBV infection and that the majority of survivor cells were ciliated-like cells located in the proximal airways. In addition to their survival, the unique gene expression profiles of these cells allowed significant morphological and physiological flexibility. Finally, we demonstrated that the presence of these cells, and potentially their unique transcriptional state, contributed to the maintenance of epithelial barrier function and benefited the host during recovery from viral infection.

It is currently unclear how IBV infection can induce long-term transcriptional changes in a surviving cell. It is likely that at least some of the long-term changes to these cells are the results of virally induced stress and/or host shutoff experienced during infection. Our survivor ciliated-like cells could be derived from either the loss of full ciliated cell identity after infection of a mature ciliated cell, or the infection of a progenitor cell that prevents complete differentiation. The stability of the ciliated cell component of the population of survivor cells and lack of BrdU labeling of ciliated survivor cells (Supplementary Fig. 6), while not definitive, is consistent with virally induced loss of phenotype in mature ciliated cells as opposed to affecting the differentiation of progenitor cells. Viewed through this framework, viral infection could induce reprogramming of ciliated cells by preventing active maintenance of ciliated cell gene expression and allowing the cell to enter an incompletely-differentiated state. The implication of this model, however, is that the ciliated cell identity is fluid in response to injury, which is a controversial framework that has been both supported[56,57] and refuted in the literature[30,58]. Regardless of the exact mechanism of their formation, it is clear that viral infection induces a population of ciliated-like cells

that uniquely express specific genes and acquire a phenotype not normally associated with ciliated cells.

Ciliated cells are crucial for the innate defense of the lung airways, using mechanical cilia beating to clear debris and pathogens from the airways of the lungs. In both mouse and human airways, ciliated cells are present in the large airways and orchestrate mucociliary clearance thereby preventing damage and infection. Our data suggest that some ciliated cells survive direct infection with an acute, cytopathic virus and display phenotypic alterations. While dynamic changes in ciliated cell populations are consistent with other lung injury models[30], the specificity of changes to survivor ciliated cells (and not uninfected ciliated cells from the same lung) argues that the processes reported here are a specific response to surviving direct IBV infection. Further, we found that cytoskeletal changes to survivor cells were conserved across both in vivo and in vitro studies, implying that this response is not due to factors outside of the infected cell, such as the immune system. While the tissue-level loss of cilia of respiratory cilia during and after viral infection has been previously reported by a number of groups[13,14,20], our work reveals that cilia loss is not solely the result of lytic killing of ciliated cells, but also potentially the virally induced elimination of functional cilia from ciliated cells. Whether or not these cells ever regain functionality or if they maintain their non-ciliated phenotype indefinitely is currently unknown. An important future area of research will be to define how these cells affect epithelial regeneration and if their dysfunction contributes to the impaired mucociliary clearance that is associated with bacterial secondary infection.

In order to assess the functional relevance of survivor cell populations, we used toxin depletion experiments to show that these cells contribute to the preservation of pulmonary barrier integrity during and after viral infection. While intuitive that depletion of survivor cells would increase epithelial permeability, we argue that toxin depletion simply reveals the total amount of epithelial damage, which would naturally result from IBV infection if no cells had the ability to survive and persist in the epithelium. Since we experimentally depleted survivor cells after the resolution of active viral infection (Supplementary Fig. 2C), it is likely that our depletion methodology actually minimized the magnitude of the survivor cell phenotype. Severe manifestations of influenza disease are characterized by rapid decline as soon as 3–4 days after viral infection and result in lung fluid accumulation and poor clinical outcomes[53]. It is likely that the most critical roles for survivor cells are during these early infection timepoints, and the phenotypes observed after DT ablation of survivor cells are consistent with a role for survivor cells limiting the severity of the primary viral disease.

As to which subsets of the survivor cell population mediate the barrier function activities, we were unable to directly answer this question as we do not have the tools to specifically deplete one specific survivor cell type and leave other survivor cell populations intact. Given that ciliated cells compose the majority of the total survivor cell population (Fig. 3), however, we believe that it is likely that the phenotypes associated with the depletion experiments are due to the elimination of survivor ciliated cells.

In sum, we have shown that some epithelial cells have the ability to survive IBV infection. Further, those survivor cells are fundamentally altered upon surviving and have the ability to help maintain epithelial barrier function. Survivor cell functionality adds a layer of complexity to current lung injury models, which generally assume basal stem cell proliferation and differentiation as the mechanism responsible for regenerating the epithelium and protecting the basement membrane. These data add to a growing body of research that show there are unappreciated cellular outcomes after viral infection, and that the resulting cellular

populations can significantly influence viral disease. Future studies further characterizing these populations will not only increase our understanding of the mechanisms of viral pathogenesis, but also potentially inform new strategies to combat disease.

## Methods

**Viruses**. Influenza B virus (B/Malaysia/2506/2004) expressing Cre recombinase was generated by inserting Cre (cloned from a PR/8-Cre plasmid[21]) into the PB1 segment of the Mal/04 genome, where the two genes are separated by PTV-1 self-cleavage site for co-translational separation. Our B/Malaysia/2506/2004 reverse genetic system is derived from a laboratory passaged virus and is no longer a perfect match to the reference sequences, however, they are still highly homologous; the exact sequences of each viral segment are available upon request. Mal/04-mNeon was generated by inserting the mNeon fluorescent gene[59] (cloned from PR/8-mNeon[24]) into the HA segment of the Mal/04 genome. All viruses were grown in a humidified incubator at 33 °C. Similarly, all viral growth curves and serial passaging was done in embryonated chicken eggs at 33 °C. The sequences for primers used in cloning are provided in Supplementary Table 3.

**Plaque assays and titering**. Mal/04-Cre was rescued using the pDZ plasmid system[60]. Each segment of the Mal/04 genome with the PB1 segment replaced with the PB1-Cre segment were transfected using Mirus TransIT-LT1 transfection reagent into 293T cells. Supernatants were collected, injected into and dilution purified in embryonated eggs. Finally, allantoic fluid was collected and plaqued on a confluent monolayer of MDCK cells (ATCC). After applying an agar overlay, the MDCKs were incubated for 72 h at 33 °C then fixed and stained with polyclonal serum to recognize viral proteins. MDCK cells were not monitored for mycoplasma infection.

**Time course of infection in cell culture**. To test the efficiency of Cre recombination by Mal/04-Cre, A549 (ATCC), a lung epithelial cell line, was transduced with a lox-STOP-lox-zsGreen transgenic insert. These cultures were then infected with WT Mal/04 or Mal/04-Cre at a range of MOI's. The expression of zsGreen was monitored using the ZOE Fluorescent Cell Imager (BioRad). A549 cells were not monitored for mycoplasma infection.

**Mouse lines and infections**. Wild-type C57BL/6, B6.Cg-Gt(ROSA)26Sor^tm14(CAG-tdTomato)Hze/J (tdTomato) and C57BL/6-Gt(ROSA) 26Sor^tm1(HBEGG)Awai/J (DTR) mice were purchased from The Jackson Laboratory. Foxj1^CreERt2 transgenic mice[61] were developed by Dr. Chay T. Kuo. Colonies of tdTomato homogenic, DTR homogenic and tdTomato;DTR heterogenic were maintained by the Duke Breeding Facility. Mice from 6–10weeksold were anesthetized with ketamine/xylazine and infected intranasally with 40 µL of virus diluted in pharmaceutical grade phosphate-buffered saline (PBS) at the indicated doses. Body weight was measured over the course of infection with 80% body weight designated as the humane endpoint. All experiments included 3–5 mice per sample, as indicated in the figure legend. No blinding was performed in animal experiments. Both male and female animals were used and randomly assigned to mock and infected treatment groups. All experiments involving animals were conducted in accordance with Duke University Animal Care and Use Committee.

**Flow cytometry**. Mouse lung epithelial cells were isolated from WT, tdTomato and DTR mice for differentiated cell culture. Lungs were perfused with 5 mL PBS then inflated with 1 mL Dispase (Corning) and 1 mL 1% low-melt NuSieve agarose (Lonza) in water. Lungs were then digested at room temperature in dispase for 45 min, further homogenized using a razor blade in dulbecco's modified eagle media (DMEM) containing DNase I (Sigma-Aldrich) and filtered through a 70 µm filter (Corning). Red blood cells were removing using 1x Red blood cell lysis buffer (BD Biosciences) then washed in PBS 1% Bovine serum albumin (BSA) before staining. Antibodies used for staining include: CD326/EpCam (BD Biosciences, clone G8.8, cat 563478, 1:100), CD45 (BD Biosciences, clone 30-F11, cat 557659, 1:100), CD31 (BD Biosciences, clone 390, cat 740690, 1:200), CD24 (BD Biosciences, clone M1/69, cat 562360, 1:75), SSEA-1 (Biolegend, clone MC-480, cat 125611, 1:50), Iso-lectin B4 (Enzo, cat ALX-65010–14001B, 1:1000) (Supplementary Table 4). Cells were incubated in primary antibodies for 1 h on ice and Live/Dead fixable blue dye (Life Technologies). Cells were analyzed on a FACSCanto II or a Fortessa X-20 (BD) machine with standard laser and filter combinations. Data was visualized and processed with FlowJo software.

**BrdU in vivo labeling**. Mice were infected and monitored until 10 days post-infection. At this point, mice were treated with BrdU (Sigma) intraperitoneally injected with 1 mg/200 µL once, and through their water at 0.8 mg/mL on days 10–14 post-infection. At 14 days, post-infection mice were collected and lung cells were analyzed using flow cytometry with an antibody staining for intracellular BrdU antibody (Life Technologies, clone MoBU-1, cat B35129).

**Differentiated cell cultures**. Basal stem cells were isolated from the tracheas of WT, tdTomato, DTR, and tdTomato;DTR mice. Tracheas were dissected, minced and digested in Pronase (Sigma) according to published protocols[62]. Fibroblasts were eliminated using differential adherence on Primaria plates (Corning) and cells were seeded on 4 µm pore transwell membranes (Corning) in mTec Plus media with Y27632 (Tocris) for 48 h before differentiation in mTec Serum-free media. Differentiation was followed by visualizing beating cilia in real time after 8 days in air–liquid interface culture. Membranes were infected between 14–21 days post-differentiation. At these timepoints, there were approximately equal numbers of apical ciliated and secretory cells.

**Microscopy**. Mouse lung tissues (Fig. 2) were fixed for a day in 2% formaldehyde at 4 °C. The tissues were then embedded in Tissue-Tek OCT (optimum cutting temperature) and cryosectioned. Mouse lung tissue microscopy was done using the 40x dry objective on a Leica AF6000 LX microscope and all largescale images were stitched together using the tilescan function.

For membranes, whole-mount tissue sections and sectioned trachea microscopy (Figs. 3–6) mice were infected as indicated in the figure legends then collected for processing. For whole-mount imaging, we collected tracheas from infected mice, opened and flattened the tissue, and imaged the epithelium from the apical side as a whole mount. All cross-sectioned tissue was embedded in OCT then cryosectioned using 7 µm sections. Samples were then washed, fixed with 2% formaldehyde at room temperature (RT), and antigen retrieval was performed in a citrate buffer or EDTA at 95 °C for 10 min before blocking in 2% BSA at RT. Samples were then stained with Actub (Abcam, clone EPR16772, cat ab179484, 1:1000), Krt5 (Biolegend, clone Poly19055, cat 905501, 1:500), FoxJ1 (eBioscience, clone 2A5, cat 14-9965, 1:500), SSEA-1 (Biolegend, clone MC-480, cat 125611, 1:250) ZO-1 (ThermoFisher, clone ZO-1-1A12, cat 33-9100, 1:500), CD24 (Abcam, clone M1/69, cat ab64064, 1:250) (Supplementary Table 4). To determine infection, samples were stained for viral antigens using sera from previously Mal/04 infected mice (1:1000). Samples were then incubated with Hoechst 33342 stain (5 µg/mL of PBS, Life Technologies H3570) to stain nuclei. Samples were mounted in Prolong Diamond (Life Technologies) and imaging was performed on either using the 40x oil SP5 inverted confocal microscope (Leica) or the ZOE Fluorescent Cell Imager (BioRad). Images were processed with Fiji (NIH).

**Foxj1 brightness quantification**. Differentiated membranes from tdTomato mice were infected with Mal/04-mNeon. At 2 DPI, membranes were fixed and stained FoxJ1 (eBioscience, clone 2A5, cat 14–9965). Samples were imaged at the same time and using consistent microscope settings on a Leica SP5 inverted confocal microscope and processed identically. The brightness of FOXJ1 in the nucleus was quantified for all cells within five images. Cells were then marked as infected or uninfected based on virally expressed mNeon and the data was normalized to uninfected cells.

**Cell surface area quantification**. The surface area of survivor cells was quantified using ALI culture membranes and in vivo. tdTomato transgenic mice (Fig. 6) or differentiated cell cultures (Supplementary Fig. 8) were infected with Mal/04-Cre and collected after 9 DPI. For whole-mount imaging of tracheal tissue, we collected tracheas from infected mice, opened and flattened the tissue, and imaged the epithelium from the apical side as a whole mount. Samples were then fixed and stained with ZO-1 (ThermoFisher, clone ZO-1-1A12, cat 33-9100, 1:500). Samples were imaged using a Leica SP5 inverted confocal microscope. Image analysis was done using Fiji to trace and quantify cell boundaries as marked by ZO-1 staining. At least three tdTomato + (survivor) and at least ten tdTomato- (uninfected) cells were quantified in each image. The cell surface area was averaged for each image and plotted as a single sample.

**Cell surface cilia quantification**. tdTomato transgenic mice were infected with Mal/04-Cre and collected at 14 DPI. Tracheas were then fixed and stained with Actub (Abcam, clone EPR16772, cat ab179484, 1:1000) and FoxJ1 (eBioscience, clone 2A5, cat 14–9965, 1:500). Survivor cells were identified based on tdTomato expression and cells were manually scored for the presence of FOXJ1 and ACTUB in both survivor cells and uninfected cells.

**10x genomics sample preparation for next-generation sequencing**. Epithelial cells were sorted (BD DiVa) using EpCam and CD45 using expression of tdTomato as the marker for infection in the 2 DPI timepoint sample. Cells were washed and libraries were prepped from 5000 cells using Chromium Single-Cell 3′ Reagent Kit (10x Genomics) by the Duke University Molecular Genomics Core. Based on the 10x Genomics Chromium literature, there is a multiplet rate of 1.6% cells/sample[63]. Libraries were then sequenced over two lanes of Illumina HiSeq 4000 with 150 bp paired end reads. The raw and normalized data is accessible at NCBI GEO under the accession number GSE116032. The reads were then processed according to 10x Genomics Cell Ranger pipeline. Samples were demultiplexed and converted to FASTQ files, then mapped against the mouse transcriptome (mm10) and the B/Mal/04 viral genome. Finally, samples were aggregated to normalize samples to the same sequencing depth. After the cell ranger pipeline, there were 7852 cells with an average of 41,405 reads per cell, a median 7165 UMI counts per cell and a

median 1986 genes per cell. This count matrix was then analyzed using R and the R package Seurat[64].

**Single-cell RNA-seq analysis using Seurat.** Using the basic filtering functions, genes expressed in fewer than three cells, cells that express fewer than 200 genes, and cells expressing >5000 genes were removed from the analysis to minimize doublets. This resulted in a total of 7833 cells for further analysis. The data matrix was then scaled to normalize gene expression counts for each cell by the total expression. These values were then multiplied by the default scale factor: 10,000 and log-normalized. Finally, variable genes were calculated using the average expression and dispersion for each gene across cells, which resulted in 1238 variable genes being identified. Common sources of variation can be regressed out based on Seurat's built-in function to construct linear models predicting gene expression based on noise-contributing variables. Using this function, variation caused by the number of detected molecules per cell was accounted for in the data. Next, principal component analysis was performed on the scaled and normalized data. Mock and 2 DPI samples were then combined using a canonical correlation analysis and the first 9CCs were used align the two samples. All samples were then grouped using the aligned CCA subspaces to cluster cells. Finally, the top 25 variably expressed genes for each cluster were calculated using the "bimod" likelihood-ratio test for single-cell gene expression.

**Sample preparation for next-generation sequencing.** Cells were sorted (BD DiVa) based on tdTomato, CD24, EpCam, and CD45 as indicated in the figure legend then RNA was prepped from at least (Mock: 78,000, 2 DPI: 4,100, 14 DPI SC: 44,000, 14 DPI: 100,000) cells using RNeasy Plus Mini kit (74104). Samples were then prepped using NEBNext Poly(A) messenger RNA (mRNA) Magnetic Isolation Module (NEB #E74905). NEBNext Ultra II RNA Library Prep Kit (E7770S) was used to prep libraries with NEBNext Multiplex Oligos for Illumina (E7335S, E7500S) for adaptor ligation and enrichment of libraries. Samples were analyzed on two lanes of Illumina HiSeq 2500 in rapid run mode. The raw and normalized data is accessible at NCBI GEO under the accession number GSE116032. The reads were mapped to the mouse transcriptome (mm10) and the B/Mal/04 genome using Bowtie, and samples were run on Basespace (Illumina). Finally reads were imported into DeSeq2 for analysis[65].

**RNA-seq analysis using DeSeq2.** First, technical duplicates across lanes were collapsed by summing them together. Then the library was scaled using the default DESeq function, which smooths over outlier read counts by log transforming read counts and puts emphasis on moderately expressed genes for normalization. Variance estimates for genes across samples were calculated and ordered and the top 200 genes were selected to plot in heatmaps. Finally, the fold change and Wald test p-values were calculated and corrected using the Benjamini-Hochberg procedure for each gene between the 14 DPI uninfected and the 14 DPI survivor cells. The top 500 genes ranked by p-value were sorted and the upregulated and downregulated genes were categorized by gene ontology using DAVID Functional Annotation Bioinformatics Microarray Analysis[40,41]. Finally, the normalized read counts were used to generate bar plots for genes of interest.

**Quantitative RT-PCR.** Cells were sorted (BD DiVa) based on tdTomato, CD24 and EpCam as indicated in the figure legend then RNA was prepped using an Rneasy Plus Mini kit (74104). One-step qRT-PCR was done using the EXPRESS One-Step Superscript qRT-PCR Kit and following commercial TaqMan probes: *SftpA* Mm00499170_m1, *SftpD* Mm00517321_m1, *Cldn18* Mm00517321_m1, *Itga7* Mm00434400_m1, and *Foxj1* Mm0126729_m1. RNA was normalized to the endogenous 18S primer probe set (Thermo 4319413E).

**Epithelial permeability.** The effect of survivor cells on epithelial permeability was tested using ALI culture membranes (Supplementary Fig. 9) and in vivo (Fig. 6). Differentiated membranes were infected with Mal/04-Cre such that there were actively infected membranes (2 DPI) and membranes, which were clear of virus (6 DPI). Membranes were then treated with 50 ng of diphtheria toxin for 2 h at 37 °C. Lastly, 250 µL of 2 mg/mL FITC-4kDa Dextran (Sigma 46944) was added to the apical membrane and the 600 µL of Hank's balanced salt solution (HBSS) was added to the basolateral side. One-hundred microliters of basolateral media was collected, diluted with HBSS then read on a plate reader in triplicate.

Similarly, the permeability of tracheal tissue after Mal/04 infection was tested by infecting both WT and DTR-transgenic mice. At 9 and 10 DPI, after virus clearance in vivo, mice were treated with diphtheria toxin both intranasally (10 ng) and intraperitoneally (100 ng). Lastly, mice were treated intranasally with 10 mg/mL FITC-4kDa Dextran. One hour later, mice were sacrificed and tracheas were collected, rinsed thoroughly with PBS and incubated in formamide before quantifying fluorescence intensity on a plate reader in duplicate. Fluorescence values were normalized to the weight of the trachea digested.

**Lung albumin ELISA.** Leakage of albumin into the lung was measured using bronchoalveolar lavage fluid from WT mice that were mock infected with PBS, WT mice that were infected with Mal/04-Cre and DTR-expressing transgenic mice that were infected with Mal/04-Cre. Both infected groups were treated with diphtheria toxin on 9 and 10 DPI both intranasally (10 ng) and intraperitoneally (100 ng) and bronchoalveolar lavage fluid was collected on day 11 post-infection. Albumin was quantified by ELISA (Bethyl E90–134) and absorbance was measured on a plate reader and compared to a standard curve.

**Forced oscillation lung mechanical measurements.** To assess lung physiology, mice were anesthetized by i.p. injection of urethane (1 g/kg), and the trachea dissected free of surrounding tissue and cannulated with a blunted 19-gauge needle. The mice were then connected to a SciReq flexiVent FX small rodent ventilator. The ventilator was programmed to maintain a positive end expiratory pressure (PEEP) of 3 cm $H_2O$ and ventilation was initiated at 150 breaths/min, tidal volume 10 mL/kg). Mice were paralyzed with i.p. pancuronium bromide (0.8 mg/kg). An electrocardiogram (ECG) monitor was connected via copper electrodes inserted subcutaneously into the limbs of the mice to determine heart rate as a monitor of depth of anesthesia. The animals were stabilized for about 5 min of regular ventilation and a standard lung volume history established by delivering two total lung capacity (TLC) maneuvers to a pressure limit of 27 cmH2O and holding for three seconds. Next, two sets of baseline measurements were collected with Snapshot and Quickprime perturbations in addition to PV-loops. Data were fit to the single-compartment model to provide values for compliance.

**Mouse morbidity measurements.** Mouse morbidity was measured in both tdTomato and DTR-transgenic mice to measure the effect of survivor cells. At 9 and 10 DPI, mice were treated with diphtheria toxin both intranasally (10 ng) and intraperitoneally (100 ng). Mice that were not recovering at 9 DPI, prior to when DT was administered, were excluded from analysis.

**Statistical analysis.** Unless otherwise noted, all statistical analysis was performed using either a two-tailed Student's T-test for tests involving one comparison between two samples. For comparisons between > 2 samples, a one-way or two-way ANOVA were used to account for numbers of independent variables. These ANOVAs were paired for different timepoints when appropriate. Fisher's least significant difference (LSD) test was used for multiple comparisons testing and S.E. M was plotted to indicate variability around the mean for tested samples[66]. All experiments included at least three biological replicates as indicated in the figure legend and were replicated at least twice. Sample sizes were chosen based on the expected magnitude and variance of the phenotype and most experiments included 3–6 samples per treatment group. No samples were excluded unless specifically stated in the figure legend and methods. Analysis was performed using Prism 7 software (Graphpad). For all figures, *$p \leq 0.05$, **$p \leq 0.001$, ns not significant.

**Reporting summary.** Further information on experimental design is available in the Nature Research Reporting Summary linked to this article.

## Data availability
The raw and analyzed data for both single-cell RNA-sequencing and bulk RNA-sequencing is accessible at NCBI GEO under the accession number GSE116032. All other data and reagents are available from the corresponding author at request.

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

## Acknowledgements

We would like to acknowledge assistance from Mike Cook and the Duke Cancer Institute Flow Cytometry Core. The high-resolution microscopy was performed at the Duke Light Microscopy Core Facility with assistance from Dr. Yasheng Gao. Lung physiology measurements were performed using Duke Rodent Inhalation Core by Barbara Theriot

and Dr. Loretta Que. The 10x Genomics pipeline for single-cell RNA-sequencing was performed at the Duke University Molecular Genomics Core by Emily Grass and Karen Abramson. Next-generation sequencing was done at the Duke Center for Genomic and Computational Biology. Sequencing reads were mapped to the mouse transcriptome and the influenza genome using applications developed by David Sachs. We thank Khadar Abdi for his contributions to the FoxJ1-CreER mouse line development. We would also like to thank Brook Heaton for cloning the PB1-Cre segment and Alfred Harding for rescue of the virus as well as Brook Heaton, Griffin Haas, and Heather Froggatt for critical reading of the manuscript. N.S.H. is partially supported by NIH 1K22-AI116509-01, 1R21-AI133444-01, 1R01-HL142985-01 and the Duke School of Medicine Whitehead Scholarship. R.E.D. is supported by the NIH Training grant T32-GM007184-41. R.A.L. is partially supported by NIH 1K22-AI110581 and R01 AI-132962. J.F.K. is supported by the NIH Training grant T32-HL007741. C.T.K. is supported by NIH 5R01-NS078192-07.

## Author contributions

R.E.D. and N.S.H. designed the study. R.E.D. performed the virus characterization, mouse infections, flow cytometry, whole-mount microscopy, sectioned microscopy, sequencing analysis, differentiated cell culture experiments, and in vivo lung function experiments. R.A.L and B.M.W. performed the experiments in Fig. 2f. R.A.L and J.K.F performed the experiments in Supplementary Fig. 6B. C.T.K provided the FoxJ1-CreER mouse line and technical guidance for the experiments in Supplementary Fig. 5B–D. R.E.D and N.S.H performed data analysis and interpretation. R.E.D. and N.S.H. wrote the manuscript.

## Additional information

**Competing interests:** The authors declare no competing interests.

