## [Peer Review File · Nature Communications]

Reviewers' Comments:

Reviewer #1:

Remarks to the Author:

The paper entitled "Ciliated cell identity loss preserves lung function during influenza B virus infection" by Heaton and coworkers describes mechanisms of Influenza B virus pathogenesis. By using a transgenic IBV strain, the authors have been capable to activate a host reporter to permanently label all infected cells. The paper contains some information of interest, which may clarify some differences existing between IAV and IBV. Overall, this study contains potentially interesting observations regarding the mechanisms by which IBV mediates its effects and a possible modulation by the host response. I am however puzzled by the fact that IBV strains can be as detrimental as IAV strains during infection. The question is therefore whether the observed effects can be related to known differences among IBV strains infectivity?

The author noticed that a specific population of cells can survive to direct IBV infection and may affect viral pathogenesis. Their demonstration is driven by the use of a transgenic IBV strain. With that approach, they identified *in vivo* and *in vitro* a population of survivor cells that they qualify as ciliated, based on the expression of markers. At the same time, they also noticed that infection alters their transcriptional profile, leading to a loss of their canonical ciliated cell identity.

Figure 2 quantifies the percentage of cells that express the reporter in EpCam⁺ or CD45⁺ positive cells. Figure 2D clearly shows that in uninfected cells, the total of these 2 populations is less than 50%. Why limit in that case to only these 2 cell types? Fig2D may suggest that there is a lot of other cell types in uninfected cells, and it might be important to better characterize them. It is important because it seems that there is also a small fraction of EpCam⁻,CD45⁻ cells that are alive after infection. If this population is still able to proliferate, it could be the origin of the permanently labeled cells.

The authors then demonstrate that infected cells are not expressing SCGB1A1, and thus are not club cells. The high numbers of survivor cells in the upper airways and the lack of SCGB1A1 staining of IBV survivor cells make convincing evidence that the cell population surviving IBV infection differs from the population surviving IAV infection (i.e. club cells).

In Figure 3, they analyzed at 14DPI the expression of basal cells (with GSIB4 lectin), secretory cells (SSEA1 labelling), and ciliated cells (CD24 labelling). The differences existing between the percentages of secretory and ciliated cells do not appear to be significant, and Figure 3D is not a conclusive demonstration of the ciliated characteristic of tdTomato⁺ cells, despite lines 159-160 that claim: "Consistent with our flow cytometry results, we found ciliated cells with mature cilia on their apical surface (Fig. 3D)", since it shows at least one cell that is "red-only" (i.e. does not express acetylated tubulin). The use of CD24 as a marker of ciliated cells is also not obvious to me, and more conventional markers of ciliated cells, such as a nuclear staining of Foxj1, should definitely be included.

The rationale in Figures 3 E-F is totally unclear for me. The single cell analysis shown in Figure 3 "utilized single-cell RNA sequencing of murine lung epithelial cells to identify markers upregulated in both uninfected cells and infected cells at peak viral replication, 2 DPI (SFig. 2)". Cells to be analyzed were positive for EpCam and negative for CD45, and came from either Mock or 2DPI B/Mal04-Cre mouse lungs. Unfortunately, no information about the differential profiles between the two backgrounds is clearly shown, and no information about the origin of the different cells that are presented is available. For me, the heatmap shown on figure 3E just represents a standard profile of mouse lung epithelial cells, and does not indicate transcripts that are differentially expressed between the 2 conditions.

The characterization of the differentially expressed transcripts shown in figure 4 should be improved: the volcano plot in Figure 4F is totally unusual; the GO enrichment are only briefly indicated on the right of the figure and poorly explained. But the decreased expression for transcripts associated with cilia and the enrichment of genes associated with cell adhesion, spreading and migration appear

intriguing. These gene expression changes after surviving B/Mal04 infection suggest that cells were potentially losing their ciliated cell transcriptional identity and responding to viral infection by changing their shape and interactions with neighboring cells. A live imaging of surviving cells may help document the observed phenotype. Such an experiment could be easily performed on their polarized primary cultures (see Figure 5 and suppl. Figure 3).

Figure 5. The description of the culture has to be improved. For instance, at which time of differentiation the cells were infected? What were at that time the different percentages of basal, ciliated and secretory cells?

Figure 6 describes several experiments using depletion experiments that suggest that the survivor cells maintain barrier integrity and preserve lung function during IBV infection.

At this point, I feel that a better justification of many observations is really needed before any editorial decision can be made. I was also worried by the fact that no information was provided about the submission of the experiments to a public repository. For me, this submission is mandatory before considering any publication. This is relevant for the single cell experiments, but also the bulk gene expression experiments.

Minor: in the different figures, a consistent labeling of the experiments should be used (e.g. figures 4c-d) to ensure an easier reading of the manuscript.

Reviewer #2:

Remarks to the Author:

General

This manuscript continues this investigator's story about epithelial cells that "survive" after influenza virus infection, in this case ciliated epithelial cells after influenza B virus infection. The manuscript needs improvements as outlined below to justify its conclusions. But even when that is done, the authors have not assigned specific gene function beyond screening RNAseq, so the title is overreaching and should more accurately reflect that at best they have found that the loss of infected lung cells causes increases in lung epithelial permeability.

Specific

1. Fig. 1. The authors claim a similar effect of wildtype and Cre-virus, but actually the data show marked attenuation of the Cre-virus in vivo (Fig 1H).

2. Fig. 2. Tomato+ cells are not evaluated as being ciliated cells by staining or other protein approach at this time point (?14 dpi) or other time points.

3. Fig. 3. Per above, CD24 as a ciliated cell marker is not fully validated at the protein level, and this protein is expressed on a wide variety of cell types. Similarly, SSEA-1 or GSIB4 are not specific for any airway epithelial cell subtype. The presence of ISG expression does not mean that a cell is infected. The RNAseq screen does not validate CD24 at the protein level. Indeed, there is no validation of any of the RNA findings, rendering this analysis (as for the staining) of uncertain significance.

4. Fig. 4. Given the above drawbacks, the variable expression is difficult to interpret. Moreover, no protein validation is provided for any of the hits.

5. Fig. 5. 5A needs to define the arrows. The in vitro function of survivors relies on increased permeability after diphtheria toxin treatment so there needs to be a control for treatment without infection. The biggest effect on permeability is at 2 days, when the infection is maximal and there are

no "survivors". So the significance of the survivor issue is uncertain. This data would appear to indicate simply that killing epithelial cells with diphtheria toxin leads to a leaky epithelium.

6. Fig. 6. The day of infection is not specified for A-D? The specificity of the functional effect being attributed to ciliated cells is uncertain based on all of the above caveats and again here no protein validation that the tomato is restricted to ciliated cells. In addition, there need to be controls for diphtheria toxin treatment of wildtype mice with and without virus. It's unlikely there is no weight change with infection compared to mock. It's also unclear why mock weight is changing with time or why it would change at all compared to its own baseline at 1 DPI. Since a mouse lung weighs less than a gram, it's also unclear what this scale means. It seems unlikely that a selective leak of ciliated epithelium would change lung weight, and indeed the changes are small. Correlations with histopathology are needed to address all of the above. In general, the data needs to be reinterpreted that loss of these cells is detrimental, rather than their presence is protective.

Reviewer #3:

Remarks to the Author:

This manuscript by Dumm et al is focused on the fate of mouse airway epithelial cells after infection by influenza virus B (IBV). Using a novel recombinant IBV which allows tagging of cells that were once infected but survive after virus clearance, the authors conclude a specific epithelial cell-type survives IBV infection and these survivor cells provide a protective function to the airways, namely maintaining epithelial barrier integrity.

Although this concept is not novel in of itself, as the senior authors have previously published this strategy with similar influenza A virus model, this new study indicates that survivor cells after IBV are ciliated epithelial cells as opposed to (predominately) Club epithelial cells as previously demonstrated for influenza A virus. The experiments are performed well and the authors use state-of-the-art reagents and techniques to demonstrate that ciliated cells are morphologically/genetically altered after survival of IBV infection.

Overall, the manuscript is well written and presented. For the most part the conclusions fit with the experimental data. Experiments describing how ciliated cells were identified as the majority of the surviving cells including morphological and genetic analysis are well described and performed. The weakest component of the manuscript relates to how ciliated cell survival impacts barrier epithelial integrity as it is difficult to appreciate the scale of the experimental changes shown in the manuscript (see below for more details).

While any study using influenza virus in mice can be criticized based on the use of one strain of virus and one strain of mouse, these studies use somewhat novel virus and mouse reagents to perform their experiments. Whether such consequences of infection are broad across influenza virus B strains or have relevance to the human infection are not addressed in this paper and are beyond the scope of this current manuscript. The manuscript does have some areas of concern which require strengthening. These concerns are both major and minor and are listed below as Overall Comments or in order of appearance in the manuscript (using line numbers for reference).

Overall Comments (Major).

1. The manuscript often refers to the distribution of epithelial cell-types in the airways and in particular, Club cells. It is not clear from the descriptions given whether the authors are referring to mouse airways or human airways in these descriptions. This should be clarified.
2. Previously, the authors showed Club cells were the (predominant) survivor cells after IAV infection while in this new manuscript, ciliated cells are the (predominant) survivors after IBV infection. Do these two viruses have different tropism for ciliated/Club cells in the mouse? This would be an important baseline finding for the interpretation of these studies.

3. The identification of ciliated cells as survivor cells is based on cellular and genetic markers. The image shown for CC10 staining in Figure 2E is worrisome as the CC10 staining appears abnormal. Is it possible that IBV infection is causing depletion of CC10 from the Club cells so that surviving Club cells are under-represented in the survivor cell pool?
4. There are some concerns overall about the assumptions that all virus infection has been cleared at the time of analysis for survivors especially in vivo. Were plaque assays performed on isolated trachea/whole lungs/tissue culture inserts at the times measurements of experimental outcomes were performed?
5. Some discussion of what the infected cells have survived through is required. This might be different in the in vitro and in vivo studies. There is no discussion of whether IBV clearance is related to cytopathic effects or immune cell-mediated clearance mechanisms and how these may be related to the in vitro and in vivo studies.
6. As noted above, there are some concerns with the Dextran/lung function studies. These concerns largely relate to the scalability of the results as it is difficult to determine the degree to which these effects are happening. The use of the DT mouse is great but other studies demonstrating what the maximal permeability of dextran through membranes without cells (given the large concentration of dextran added) would be informative. Why was 4K Dextran chosen? Additionally, the in vivo dextran measures are hard to interpret as it is not clear how much dextran reached the trachea and where that dextran may have ended up in (presumably) the trachea. Morphologic studies may be informative in this case. Finally, although lung function studies (compliance) are presented, it is unclear what scale of lung function loss these studies represent. Can these be put into context with other treatments that affect lung compliance.

Specific Comments (Major and Minor)

Line 1 (major): This is a study in mouse airways and the title should reflect that fact. Other comments in the manuscript should also emphasize these are exclusively mouse studies.

Line 1 (minor): The title is difficult to immediately appreciate...it is not apparent to the reader what "ciliated cell identity" really means.

Line 77 (minor): "some cells"...the specific cell-types should be identified here.

Line 82 (minor): "affect lung physiology" is a broad statement. The study describes a single experiment measure of lung function i.e., lung compliance.

Line 95 (minor): alternative to SOS please.

Line 128 (minor): "4% of cells"...is this total lung cells or specifically epithelial cells?

Line 318 (minor): There remain many unknowns for almost all diseases, please remove this overly dramatic sentence.

Line 326 (minor): Upper airways most commonly refers to the nasal/oropharyngeal airway regions...not the trachea.

Line 368 (minor): Is the data supporting the claim that Club cells are also survivors shown?

Discussion: Some discussion should be made about the relative distributions of ciliated/Club cells between the mouse and the human airways to show relevance to the human disease. Also, some discussion of how basal cells have commonly been thought to assume the role of cells that provide protection after influenza virus induced barrier disruption should be included.

Figure 1: The recombinant CRE virus consistently shows lower titers. Are there any potential reasons for this?

Figure 2E: CC10 staining appears depleted.

Figure 5A/B: Is it possible that IBV infection is causing FoxJ1 to aggregate in the infected cells so giving the impression that brightness is decreased.

Figure 5C: These "from the top views" are unconvincing. Cross-sectional morphology is required to understand how the morphology of the survivor cells is being altered.

Dumm et al., response to reviewers:

We thank reviewers for their thoughtful critique of the manuscript. We have addressed concerns individually below and highlighted changes to the main text with underlining in that document.

Reviewer #1 (Remarks to the Author):

The paper entitled “Ciliated cell identity loss preserves lung function during influenza B virus infection” by Heaton and coworkers describes mechanisms of Influenza B virus pathogenesis. By using a transgenic IBV strain, the authors have been capable to activate a host reporter to permanently label all infected cells. The paper contains some information of interest, which may clarify some differences existing between IAV and IBV. Overall, this study contains potentially interesting observations regarding the mechanisms by which IBV mediates its effects and a possible modulation by the host response. I am however puzzled by the fact that IBV strains can be as detrimental as IAV strains during infection. The question is therefore whether the observed effects can be related to known differences among IBV strains infectivity? The author noticed that a specific population of cells can survive to direct IBV infection and may affect viral pathogenesis. Their demonstration is driven by the use of a transgenic IBV strain. With that approach, they identified in vivo and in vitro a population of survivor cells that they qualify as ciliated, based on the expression of markers. At the same time, they also noticed that infection alters their transcriptional profile, leading to a loss of their canonical ciliated cell identity.

1. Figure 2 quantifies the percentage of cells that express the reporter in EpCam+ or CD45+ positive cells. Figure 2D clearly shows that in uninfected cells, the total of these 2 populations is less than 50%. Why limit in that case to only these 2 cell types?

In the original experiment, we had poor lysis of red blood cells which were appearing as double negative cells. We have replaced Fig 2C,D with a cleaner preparation to better represent the relative lung cell populations.

2. Fig2D may suggest that there is a lot of other cell types in uninfected cells, and it might be important to better characterize them. It is important because it seems that there is also a small fraction of EpCam-,CD45- cells that are alive after infection. If this population is still able to proliferate, it could be the origin of the permanently labeled cells.

We agree that there are likely different cell types (endothelial, fibroblasts) present in the lung, however we found that over 90% of survivor cells were epithelial (Fig 2C). We are also forced to be cautious in our interpretation of the marker-negative cells as the enzymatic digestion in the sample preparation which could account for cell-surface marker loss.

The authors then demonstrate that infected cells are not expressing SCGB1A1, and thus are not club cells. The high numbers of survivor cells in the upper airways

and the lack of SCGB1A1 staining of IBV survivor cells make convincing evidence that the cell population surviving IBV infection differs from the population surviving IAV infection (i.e. club cells).

3. In Figure 3, they analyzed at 14DPI the expression of basal cells (with GSIB4 lectin), secretory cells (SSEA1 labelling), and ciliated cells (CD24 labelling). The differences existing between the percentages of secretory and ciliated cells do not appear to be significant, and Figure 3D is not a conclusive demonstration of the ciliated characteristic of tdTomato+ cells, despite lines 159-160 that claim: “Consistent with our flow cytometry results, we found ciliated cells with mature cilia on their apical surface (Fig. 3D)”, since it shows at least one cell that is “red-only” (i.e. does not express acetylated tubulin).

We agree that flow cytometry is not fully diagnostic of cell type. To validate the flow cytometry results, we added murine tracheal tissue sectioning of survivor cells stained with Krt5, SSEA1, and FOXJ1 as a more conclusive demonstration of the ciliated characteristic of tdTomato+ survivor cells (Fig 3E,F).

4. The use of CD24 as a marker of ciliated cells is also not obvious to me, and more conventional markers of ciliated cells, such as a nuclear staining of Foxj1, should definitely be included.

We have added microscopy of survivor cells in murine proximal airways stained for nuclear FOXJ1 expression (Fig 3F). However, because intracellular FoxJ1 staining would not work for our RNAseq analysis, we still relied on CD24 for flow cytometry. Because of this, we have added extensive characterization of CD24 as a marker for ciliated cells using single-cell sequencing and qRT-PCR (Supplemental Fig 2).

5. The rationale in Figures 3 E-F is totally unclear for me. The single cell analysis shown in Figure 3 “utilized single-cell RNA sequencing of murine lung epithelial cells to identify markers upregulated in both uninfected cells and infected cells at peak viral replication, 2 DPI (SFig. 2)”. Cells to be analyzed were positive for EpCam and negative for CD45, and came from either Mock or 2DPI B/Mal04-Cre mouse lungs. Unfortunately, no information about the differential profiles between the two backgrounds is clearly shown, and no information about the origin of the different cells that are presented is available. For me, the heatmap shown on figure 3E just represents a standard profile of mouse lung epithelial cells, and does not indicate transcripts that are differentially expressed between the 2 conditions.

We have clarified our use of single-cell sequencing to define appropriate markers of ciliated cells in the lung before and after infection in Supplemental Fig 2. Specifically, we collected total EpCam+ cells from uninfected mice and performed unbiased clustering of the different lung epithelial cells (Supplemental Fig 3A-C). We used this data to show the selectivity of CD24 for ciliated cells compared to other cell types (Supplemental Fig 3D). Finally, we verified the maintenance of CD24 expression on infected cells (Supplemental Fig 3D-F).

6. The characterization of the differentially expressed transcripts shown in figure 4 should be improved: the volcano plot in Figure 4F is totally unusual; the GO enrichment are only briefly indicated on the right of the figure and poorly explained. But the decreased expression for transcripts associated with cilia and the enrichment of genes associated with cell adhesion, spreading and migration appear intriguing.

We have changed the presentation and explanation of the GO enrichment analysis to better highlight cellular processes up and downregulated in survivor ciliated cells (Fig 4F).

7. These gene expression changes after surviving B/Mal04 infection suggest that cells were potentially losing their ciliated cell transcriptional identity and responding to viral infection by changing their shape and interactions with neighboring cells. A live imaging of surviving cells may help document the observed phenotype. Such an experiment could be easily performed on their polarized primary cultures (see Figure 5 and suppl. Figure 3).

Due to technical limitations, we were unable to use live imaging to observe the change in morphology. To address this point however, we added an in-depth characterization of the phenotypic changes (including cilia loss) to survivor cells (Fig 5).

8. Figure 5. The description of the culture has to be improved. For instance, at which time of differentiation the cells were infected? What were at that time the different percentages of basal, ciliated and secretory cells?

A more detailed description of the ALI culture has been added in the Supplemental Fig 4 Legend as well as in the Materials & Methods.

9. Figure 6 describes several experiments using depletion experiments that suggest that the survivor cells maintain barrier integrity and preserve lung function during IBV infection. At this point, I feel that a better justification of many observations is really needed before any editorial decision can be made.

We have clarified our experimental designs for the assays utilized in figure 6 and rewritten the text for more clarity.

10. I was also worried by the fact that no information was provided about the submission of the experiments to a public repository. For me, this submission is mandatory before considering any publication. This is relevant for the single cell experiments, but also the bulk gene expression experiments.

We have deposited the raw data at NCBI Geo and have listed the accession numbers in the Materials & Methods under Data Availability.

11. Minor: in the different figures, a consistent labeling of the experiments should be used (e.g. figures 4c-d) to ensure an easier reading of the manuscript.

We have clarified labeling of the panels in Figures 4 and labels throughout the paper.

Reviewer #2 (Remarks to the Author):

General

This manuscript continues this investigator's story about epithelial cells that "survive" after influenza virus infection, in this case ciliated epithelial cells after influenza B virus infection. The manuscript needs improvements as outlined below to justify its conclusions.

12. But even when that is done, the authors have not assigned specific gene function beyond screening RNAseq, so the title is overreaching and should more accurately reflect that at best they have found that the loss of infected lung cells causes increases in lung epithelial permeability.

We have changed the title to more accurately reflect the conclusions of the manuscript.

Specific

13. Fig. 1. The authors claim a similar effect of wildtype and Cre-virus, but actually the data show marked attenuation of the Cre-virus in vivo (Fig 1H).

We have clarified the extent of attenuation of the Mal04-Cre virus compared to wildtype in the text and included references to the attenuation previously reported with other influenza reporter viruses (lines 134-139).

14. Fig. 2. Tomato+ cells are not evaluated as being ciliated cells by staining or other protein approach at this time point (?14 dpi) or other time points.

We have added murine tracheal tissue sectioning of survivor cells stained with FOXJ1 as a more conclusive demonstration of the ciliated characteristic of tdTomato+ survivor cells (Fig 3F).

15. Fig. 3. Per above, CD24 as a ciliated cell marker is not fully validated at the protein level, and this protein is expressed on a wide variety of cell types.

We have added immunofluorescence data staining for FOXJ1 protein in survivor cells, validating that survivor cells are predominantly ciliated cells. Since technically we were forced to rely on a surface expressed marker for isolating ciliated cells in some experiments, we have also added single cell RNAseq and qRT-PCR characterization of EpCam+ cells from the lung to validate CD24 as the best flow cytometry marker available to label ciliated cells (Supplemental Fig 2).

16. Similarly, SSEA-1 or GSIB4 are not specific for any airway epithelial cell subtype.

We have clarified that our use of SSEA-1 and GSIB4 was to initially broadly define lung survivor cell subtypes (lines 176-178). To be more specific for individual cell types, we have added data using immunofluorescence for more diagnostic markers in survivor cells detected in murine tracheal tissue (Fig 3E,F).

17. The presence of ISG expression does not mean that a cell is infected.
We have replaced the ISG data with viral RNA to demonstrate infection (Supplemental Fig 2E, Fig 4B)
18. The RNAseq screen does not validate CD24 at the protein level. Indeed, there is no validation of any of the RNA findings, rendering this analysis (as for the staining) of uncertain significance. Fig. 4. Given the above drawbacks, the variable expression is difficult to interpret. Moreover, no protein validation is provided for any of the hits.
We have clarified in the text that a cell surface marker for the protein CD24 was used for cell sorting prior to RNA-sequencing analysis. Due to limited cell numbers, western blot validation of altered gene expression in survivor cell populations proved technically difficult. To address the point of RNAseq validation, we were able to validate our RNAseq predictions via qRT-PCR and immunofluorescence of FoxJ1 protein levels in infected/survivor cells (Fig 5, Supplemental Fig 3). Further, we performed validation of the phenotypic changes to survivor cells predicted by our RNAseq data with respect to loss of apical cilia (Fig 5).
19. Fig. 5. 5A needs to define the arrows.
This has been corrected in the Figure Legend.
20. The in vitro function of survivors relies on increased permeability after diphtheria toxin treatment so there needs to be a control for treatment without infection.
We have added a control for DT treatment of Mock infected membranes (Supplemental Fig 6D).
21. The biggest effect on permeability is at 2 days, when the infection is maximal and there are no “survivors”. So the significance of the survivor issue is uncertain. This data would appear to indicate simply that killing epithelial cells with diphtheria toxin leads to a leaky epithelium.
We initially included toxin depletion at 2DPI to show how much permeability would be affected if all infected cells were cleared. To better demonstrate the relevance of survivor cells, we have shifted focus of the studies to our in vivo system and better detailed our conclusions. To the second point, we agree that toxin killing of cells leads to a leaky epithelium. Our overall conclusion however, is that normally, the survival of these cells limits permeability induced by viral infection (i.e. toxin is only doing what the virus would have done if these cells did not have the ability to survive). This point is made in the discussion (lines 421-424).
22. Fig. 6. The day of infection is not specified for A-D?
This information has been added to the Figure 6 legend.

23. The specificity of the functional effect being attributed to ciliated cells is uncertain based on all of the above caveats and again here no protein validation that the tomato is restricted to ciliated cells.
We have clarified our conclusion that the phenotype is the result of the depletion of the total survivor cell population, and we cannot ascribe the barrier function phenotypes exclusively to ciliated cells (lines 437-443).
24. In addition, there need to be controls for diphtheria toxin treatment of wildtype mice with and without virus. It's unlikely there is no weight change with infection compared to mock.
We have added a control for DT treatment in Mock infected mice (Supplemental Fig 7). We have also changed the plotting of the data to make the mouse body-weight changes more apparent.
25. It's also unclear why mock weight is changing with time or why it would change at all compared to its own baseline at 1 DPI. Since a mouse lung weighs less than a gram, it's also unclear what this scale means. It seems unlikely that a selective leak of ciliated epithelium would change lung weight, and indeed the changes are small. Correlations with histopathology are needed to address all of the above.
The data recorded in Figure 6J is mouse bodyweight as a measure of morbidity of the animal. The bodyweight of the mock infected animals increases over time because the animals are not fully grown at the time of infection. We have clarified what was plotted on the graph axis as well as the Figure 6 legend.
26. In general, the data needs to be reinterpreted that loss of these cells is detrimental, rather than their presence is protective.
This change in language about the roles of survivor epithelial cells has been changed throughout the manuscript.

Reviewer #3 (Remarks to the Author):

This manuscript by Dumm et al is focused on the fate of mouse airway epithelial cells after infection by influenza virus B (IBV). Using a novel recombinant IBV which allows tagging of cells that were once infected but survive after virus clearance, the authors conclude a specific epithelial cell-type survives IBV infection and these survivor cells provide a protective function to the airways, namely maintaining epithelial barrier integrity. Although this concept is not novel in of itself, as the senior authors have previously published this strategy with similar influenza A virus model, this new study indicates that survivor cells after IBV are ciliated epithelial cells as opposed to (predominately) Club epithelial cells as previously demonstrated for influenza A virus. The experiments are performed well and the authors use state-of-the-art reagents and techniques to demonstrate that ciliated cells are morphologically/genetically altered after survival of IBV infection. Overall, the manuscript is well written and presented. For the most part the conclusions fit with the experimental data. Experiments describing how ciliated cells were identified as the majority of the surviving cells including morphological

and genetic analysis are well described and performed. The weakest component of the manuscript relates to how ciliated cell survival impacts barrier epithelial integrity as it is difficult to appreciate the scale of the experimental changes shown in the manuscript (see below for more details). While any study using influenza virus in mice can be criticized based on the use of one strain of virus and one strain of mouse, these studies use somewhat novel virus and mouse reagents to perform their experiments. Whether such consequences of infection are broad across influenza virus B strains or have relevance to the human infection are not addressed in this paper and are beyond the scope of this current manuscript. The manuscript does have some areas of concern which require strengthening. These concerns are both major and minor and are listed below as Overall Comments or in order of appearance in the manuscript (using line numbers for reference).

Overall Comments (Major).

27. The manuscript often refers to the distribution of epithelial cell-types in the airways and in particular, Club cells. It is not clear from the descriptions given whether the authors are referring to mouse airways or human airways in these descriptions. This should be clarified.

We have specified the use of murine cells and models throughout the text.

28. Previously, the authors showed Club cells were the (predominant) survivor cells after IAV infection while in this new manuscript, ciliated cells are the (predominant) survivors after IBV infection. Do these two viruses have different tropism for ciliated/Club cells in the mouse? This would be an important baseline finding for the interpretation of these studies.

We have added data showing the tropism of IBV and how that relates to the cells that survive infection (Fig 2A,B). We also felt that the manuscript was the clearest when focused on IBV alone, we have therefore removed comparisons between IAV and IBV infections.

29. The identification of ciliated cells as survivor cells is based on cellular and genetic markers. The image shown for CC10 staining in Figure 2E is worrisome as the CC10 staining appears abnormal. Is it possible that IBV infection is causing depletion of CC10 from the Club cells so that surviving Club cells are under-represented in the survivor cell pool?

We have replaced these panels with a more comprehensive staining for different cell markers in mouse lungs. Based on the presence of FOXJ1 and CD24 rather than the absence of CC10 (Fig 3F), we conclude that the majority of survivor cells are ciliated. It is certainly possible that some cells appear to be lineage negative based on viral depletion of specific markers; we therefore focused our analysis on defining characteristics of marker positive (i.e. FoxJ1+) survivor cell populations.

30. There are some concerns overall about the assumptions that all virus infection has been cleared at the time of analysis for survivors especially in vivo. Were plaque assays performed on isolated trachea/whole lungs/tissue culture inserts at the times measurements of experimental outcomes were performed?

We have added experiments measuring viral titers from isolated mouse lungs (Supplemental Fig 1C) as well as viral clearance from ALI culture inserts (Supplemental Fig 4C,D).

31. Some discussion of what the infected cells have survived through is required. This might be different in the *in vitro* and *in vivo* studies. There is no discussion of whether IBV clearance is related to cytopathic effects or immune cell-mediated clearance mechanisms and how these may be related to the *in vitro* and *in vivo* studies.

The mechanisms of how cells survive lytic viral infection are not known, and are an area of active research in our laboratory. We have added a brief discussion of this point (and how the mechanisms of survival may influence the survival cell phenotype) to the discussion (lines 402-405).

32. As noted above, there are some concerns with the Dextran/lung function studies. These concerns largely relate to the scalability of the results as it is difficult to determine the degree to which these effects are happening. The use of the DT mouse is great but other studies demonstrating what the maximal permeability of dextran through membranes without cells (given the large concentration of dextran added) would be informative.

In the revised paper, we have focused on *in vivo* measurements in the main text and moved the *in vitro* studies to the supplemental material as supportive data. Further, we have added a control of the permeability of a transwell membrane without cells present (Supplemental Fig 6D).

33. Why was 4K Dextran chosen?

We chose the 4K dextran based on a previously published study and we have cited that paper in the main text (Line 312).

34. Additionally, the *in vivo* dextran measures are hard to interpret as it is not clear how much dextran reached the trachea and where that dextran may have ended up in (presumably) the trachea. Morphologic studies may be informative in this case.

While we were unable to include morphologic studies using dextran leakage due to technical difficulties, we have included additional morphologic studies of the morphology in murine tracheas (Fig 5).

35. Finally, although lung function studies (compliance) are presented, it is unclear what scale of lung function loss these studies represent. Can these be put into context with other treatments that affect lung compliance.

More discussion of the physiological relevance of changes to lung compliance as well as citations of previous studies has been added (line 336).

Specific Comments (Major and Minor)

36. Line 1 (major): This is a study in mouse airways and the title should reflect that fact. Other comments in the manuscript should also emphasize these are exclusively mouse studies.
We have specified the use of murine cells and models in the text and title.
37. Line 1 (minor): The title is difficult to immediately appreciate...it is not apparent to the reader what “ciliated cell identity” really means.
We have changed the title to more accurately reflect the conclusions of the manuscript.
38. Line 77 (minor): “some cells”...the specific cell-types should be identified here.
Specific cell types have been added to the text.
39. Line 82 (minor): “affect lung physiology” is a broad statement. The study describes a single experiment measure of lung function i.e., lung compliance.
This has been changed to explicitly state that we measured lung compliance.
40. Line 95 (minor): alternative to SOS please.
This phrase has been removed from the text.
41. Line 128 (minor): “4% of cells”....is this total lung cells or specifically epithelial cells?
This has been clarified in the text that we are referring specifically to epithelial cells.
42. Line 318 (minor): There remain many unknowns for almost all diseases, please remove this overly dramatic sentence.
This sentence has been removed from the text.
43. Line 326 (minor): Upper airways most commonly refers to the nasal/oropharyngeal airway regions...not the trachea.
This has been corrected to “proximal airways” throughout the text.
44. Line 368 (minor): Is the data supporting the claim that Club cells are also survivors shown?
Fig 3C-E support the claim that some club cells survive, however the vast majority of survivors express markers of ciliated cells.
45. Discussion: Some discussion should be made about the relative distributions of ciliated/Club cells between the mouse and the human airways to show relevance to the human disease. Also, some discussion of how basal cells have commonly been thought to assume the role of cells that provide protection after influenza virus induced barrier disruption should be included.
A more thorough discussion of the distribution of epithelial cell types and the normal mechanisms of barrier protection after influenza virus infection has been added in the discussion.

46. Figure 1: The recombinant CRE virus consistently shows lower titers. Are there any potential reasons for this?

We have clarified the extent of attenuation of the Mal04-Cre virus compared to wildtype and previously described influenza reporter viruses (lines 137-139).

47. Figure 2E: CC10 staining appears depleted.

We replaced these panels with more comprehensive staining for different cell types of the mouse lung (Fig 3E,F) to support our conclusions.

48. Figure 5A/B: Is it possible that IBV infection is causing FoxJ1 to aggregate in the infected cells so giving the impression that brightness is decreased.

To look specifically at FoxJ1 transcription, we have added qRT-PCR to demonstrate that the RNA levels are lower in survivor cells (Fig 5A).

49. Figure 5C: These “from the top views” are unconvincing. Cross-sectional morphology is required to understand how the morphology of the survivor cells is being altered.

We have added cross-sectional staining to validate protein expression in survivor cells, however the sections are too thin to get a good quantification of total morphology. We therefore utilized flow cytometry in addition to the apical surface measurements to measure cellular morphology after infection (Fig 6A-D). Further, we have added a figure describing additional morphological alterations (such as those to apical cilia) after surviving infection (Fig 5).

Reviewers' Comments:

Reviewer #1:

Remarks to the Author:

The new version of the manuscript has clarified most of my questions, and its quality has clearly improved. I still have some concerns about some parts, though. I agree with reviewer 3 that a weak component of the manuscript relates to how ciliated cell survival impacts barrier epithelial integrity. The difference between DT and control is modest (which may be expected, considering the small contribution that can be expected from the airways relative to distal lung, which should result in modest impact on more global parameters such as the BAL content). There are certainly several possible interpretations to their data, besides a protection by survivor cells. Although interesting, the diphtheria toxin experiment does not provide very clear-cut results that would make the conclusions definitive.

Back to the identification of survivor cells, a possibility still remains that the cells that are infected correspond to a subgroup of the multiciliated precursors. According to the experimental set-up, cells would have then about ten days to differentiate into a multiciliated cell. How is it possible to rule out that the infection is indeed driving this differentiation, rather than affecting directly the well differentiated multiciliated cells?

I have also noticed during the reading of this new version that the single cell experiment was performed with a very high number of cells: after a quite stringent filter, the authors still have more than 7000 cells for their analysis. At this level, this implies probably that there are between 5 and 10% of doublet cells, and this high number may interfere with some of the interpretations. For instance, is not it an explanation for their identification of markers of the distal lung? The authors should really clarify this technical issue.

Reviewer #2:

Remarks to the Author:

General Comment

The authors added data or made text changes but didn't adequately address the comments. Remain issues:

Specific Comments

1. In Figure 1c and e, why are the authors using chicken eggs instead of a cell line? The Cre-expressing virus is clearly less virulent (Fig 1h) but viral titers look nearly identical (Fig 1e). Perhaps the discrepancy is due to the use of chicken eggs instead of a more relevant cell line such as A549 which they use for other experiments in the same figure.

2. In Figure 1d and g, why not show staining for IBV in addition to the zsGreen? This would allow confirmation that the infections look similar between viruses and that all IBV-Cre infected cells express zsGreen.

3. For Figure 3a, it would be useful to show data for time points in between 2 and 14 days. In IAV, most of the epithelial destruction is observed at 3-8 days. It would be good to show these "survivor cells" sitting alone on a denuded epithelium as opposed to before and after destruction. This comment can also be applied to Figure 4.

4. The use of CD24 was questioned by two of the three reviewers (Comments #4 and #15). The

authors provide additional RNA data but limited protein data.

5. For Figure 5a, what day post-infection is this data?

6. For Figure 6d, why compare survivor cell size only to uninfected neighbors? The neighboring cells will be different types of cells. The authors should show a comparison to mock infected cells of the same cell type.

7. As also noted by Reviewer #1 Comment #2, if EpCAM-CD45- cells are being infected and then surviving and differentiating into other cell types, this would fundamentally undercut the premise of the manuscript. The authors never really address this issue and instead just caution that their enzymatic digestion may have stripped off cell-surface markers.

Reviewer #3:

Remarks to the Author:

The authors have provided a reasonable rebuttal to my comments regarding the manuscript and as a result have alleviated many of my concerns. I am still not convinced the title represents the data presented but I do prefer the new title compared to the original. In my version of the revised document the title still does not include "in mice" which is an absolute requirement for these studies. One other point is related to the comment in the Discussion at line 389 which describes ciliated cell survival after infection. I doubt that all infected ciliated cells survive virus infection but the discussion implies that. This requires some context of what is most likely happening.

Dumm et al., response to reviewers:

We thank reviewers for their review of the manuscript and additional comments. We have added new data and clarified several points in the text for increased clarity. Detailed responses to each specific point can be found below.

Additionally, in order to access the raw sequencing data (which is deposited by not yet publicly available), please follow the instructions below for reviewer specific access:

To review GEO accession GSE116032:

Go to <https://www.ncbi.nlm.nih.gov/geo/query/acc.cgi?acc=GSE116032>

Enter token upgxmkuvczuvpad into the box

Reviewer #1 (Remarks to the Author):

The new version of the manuscript has clarified most of my questions, and its quality has clearly improved. I still have some concerns about some parts, though. I agree with reviewer 3 that a weak component of the manuscript relates to how ciliated cell survival impacts barrier epithelial integrity. The difference between DT and control is modest (which may be expected, considering the small contribution that can be expected from the airways relative to distal lung, which should result in modest impact on more global parameters such as the BAL content).

1. There are certainly several possible interpretations to their data, besides a protection by survivor cells. Although interesting, the diphtheria toxin experiment does not provide very clear-cut results that would make the conclusions definitive.

We agree with the reviewer and have changed our language to indicate that our data are consistent with our model but not definitive proof, lines 455-458.

2. Back to the identification of survivor cells, a possibility still remains that the cells that are infected correspond to a subgroup of the multiciliated precursors. According to the experimental set-up, cells would have then about ten days to differentiate into a multiciliated cell. How is it possible to rule out that the infection is indeed driving this differentiation, rather than affecting directly the well differentiated multiciliated cells?

We agree that there could be multiple sources of the abnormal ciliated cell phenotypes and have outlined potential models in the results, lines 269-281. As a starting point for distinguishing between these models, we have included additional experiments (Supplementary Fig. 5). We performed a time course of ciliated survivor cells and show their numbers (as a percentage of total survivor cells) are stable over time. To more directly assess if our abnormal ciliated cells were derived from aberrant differentiation, we also performed BrdU staining on CD24+ survivors and saw very little BrdU uptake (Supplementary Fig. 5A). We assayed proliferation as progenitor cells actively proliferate prior to differentiation (*Kumar*,

P. A. et al. Distal airway stem cells yield alveoli in vitro and during lung regeneration following H1N1 influenza infection. Cell 147, 525-538, doi:10.1016/j.cell.2011.10.001 (2011). Although we don't have the tools to conclusively resolve this question, we believe that this additional data are consistent with our the model wherein abnormal ciliated cells are derived from dedifferentiated ciliated cells. We have updated our discussion to be more precise on this point (lines 428-431).

3. I have also noticed during the reading of this new version that the single cell experiment was performed with a very high number of cells: after a quite stringent filter, the authors still have more than 7000 cells for their analysis. At this level, this implies probably that there are between 5 and 10% of doublet cells, and this high number may interfere with some of the interpretations. For instance, is not it an explanation for their identification of markers of the distal lung? The authors should really clarify this technical issue.

We agree that there are likely some doublets in our single cell analysis. In our materials and methods, we have included a description of the expected doublet rate based on the number of cells analyzed. Based on the protocol we used, the 10X Genomics literature estimates that we should expect a multiplet rate of ~1.6% (Zheng, G. X. et al. Massively parallel digital transcriptional profiling of single cells. Nat Commun 8, 14049, doi:10.1038/ncomms14049 (2017)) which is unavoidable with this technique (cited in lines 621-622).

We also attempted to further minimize multiplet issues confounding our analysis. We included a filtering step in the bioinformatic analysis where we removed cells containing greater than 5000 genes to minimize multiplets. This step is part of the 10X Genomics pipeline and its use has been clarified in the Materials and Methods (lines 634-636). While it is difficult to know how effective this step was, it likely reduced our multiplet numbers in the analysis.

To the specific scientific point however, we don't believe that multiplets are a possible explanation for detecting markers of the distal lung. In this paper, we only used single-cell RNA sequencing to validate CD24 during infection as a stable marker for ciliated cells. Additionally, we reported clusters representing cell types with relatively low resolution to represent major differences in cell types rather than rare populations of cells. And we also went on to validate CD24 as a marker for ciliated cells using other techniques (Supplementary Fig. 3). If we understand the reviewer correctly, they are referring to the detection of surfactant proteins in survivor ciliated cell populations. This analysis was not based on single-cell RNAsequencing data, but rather the bulk RNA-sequencing of the survivor ciliated cell population described in Figure 4. In addition to that sequencing experiment, we went on to verify the expression of those markers (SftpA/C/D) using non-sequencing techniques (Supplementary Fig. 4).

Reviewer #2 (Remarks to the Author):

General Comment

The authors added data or made text changes but didn't adequately address the comments.

Specific Comments

1. In Figure 1c and e, why are the authors using chicken eggs instead of a cell line? The Cre-expressing virus is clearly less virulent (Fig 1h) but viral titers look nearly identical (Fig 1e). Perhaps the discrepancy is due to the use of chicken eggs instead of a more relevant cell line such as A549 which they use for other experiments in the same figure.

We simply reported egg-based growth curves for technical ease of the experiments. Since all *in vitro* growth assays are artificial experimental systems, we favor the reported *in vivo* LD50 of the virus as the critical piece of data to understand how the introduction of Cre affected viral virulence.

2. In Figure 1d and g, why not show staining for IBV in addition to the zsGreen? This would allow confirmation that the infections look similar between viruses and that all IBV-Cre infected cells express zsGreen.

We have added the requested analysis of cells infected with Mal/04-Cre and costained for viral antigens in Supplementary Fig. 1. In this type of analysis however, not all infected cells are expected to become ZsGreen positive as high levels of viral replication induces effective host-shutoff and prevents ZsGreen expression. It is also worth pointing out that even if we had Cre-negative viruses in our animal inoculum, this would only decrease the apparent amount of survivor cells in our analysis and not affect the conclusions of the paper.

3. For Figure 3a, it would be useful to show data for time points in between 2 and 14 days. In IAV, most of the epithelial destruction is observed at 3-8 days. It would be good to show these "survivor cells" sitting alone on a denuded epithelium as opposed to before and after destruction. This comment can also be applied to Figure 4.

Because the virus is not cleared until ~9 days post-infection *in vivo*, tdTomato+ cells at earlier timepoints reflect a combination of both actively infected cells and "survivor" cells. Interpreting these experiments would be extremely difficult.

4. The use of CD24 was questioned by two of the three reviewers (Comments #4 and #15). The authors provide additional RNA data but limited protein data.

Based on previous reviews, we added Foxj1 protein staining of infected murine tracheas as validation of survivor ciliated cells. We were limited to CD24 as this was the only available surface marker of ciliated cells. In addition to FoxJ1 staining, we also performed single-cell RNA sequencing data in Supplementary Fig. 3 as well as Foxj1 expression data as validation of CD24 as a flow cytometry marker.

5. For Figure 5a, what day post-infection is this data?

This information has been added to the Figure 5 legend (lines 1063).

6. For Figure 6d, why compare survivor cell size only to uninfected neighbors? The neighboring cells will be different types of cells. The authors should show a comparison to mock infected cells of the same cell type.

We have added analysis comparing survivor cells to uninfected ciliated cells (Fig. 6E, lines 292-295).

7. As also noted by Reviewer #1 Comment #2, if EpCAM-CD45- cells are being infected and then surviving and differentiating into other cell types, this would fundamentally undercut the premise of the manuscript. The authors never really address this issue and instead just caution that their enzymatic digestion may have stripped off cell-surface markers.

Please see the detailed response to reviewer 1, comment number 2.

Reviewer #3 (Remarks to the Author):

The authors have provided a reasonable rebuttal to my comments regarding the manuscript and as a result have alleviated many of my concerns. I am still not convinced the title represents the data presented but I do prefer the new title compared to the original. In my version of the revised document the title still does not include "in mice" which is an absolute requirement for these studies. One other point is related to the comment in the Discussion at line 389 which describes ciliated cell survival after infection. I doubt that all infected ciliated cells survive virus infection but the discussion implies that. This requires some context of what is most likely happening.

We have clarified the language in the discussion (lines 400-403) that it is only a subset of infected ciliated cells that survive viral infection.

Reviewers' Comments:

Reviewer #1:

Remarks to the Author:

For me, the demonstration is still not convincing. The observation of an expression of surfactant proteins does not fit with a ciliated cell origin. The association between CD24 and ciliated cell is not at all justified: to the best of my knowledge, it is usually said that CD25 is expressed in B lymphocytes, differentiating neuroblasts, neutrophils and their precursors (see Genecards, for instance), and I did not find a clear expression in ciliated cells. So the selection of cells based on this marker makes the demonstration very confusing to me. I understand that altered expression after infection can affect the expression of classical markers of ciliated cells, but a better choice of the bait seems to me mandatory and weakening the demonstration.

I suspect that a better exploration of single cell RNA sequencing data would help clarifying the situation, even though I don't understand the use of the 2DPI condition. More information would have come from the latest time point, showing the survivors. With that in hand, it would be possible to use a program such as Monocle to build the lineage of these cells. That would certainly clarify a lot the demonstration. Unfortunately, at this point, the information provided by the single cell is not sufficient.

I agree with reviewer #1 that the diphtheria toxin experiment does not provide very clear-cut results. Overall, the study still lack the elements that would make the conclusions definitive.

Minor:

SftpA in Figure 4g is misspelled.

Moreover, page 14 probably refers to Supplementary Fig. 8C (Fig. 1C that does not exist!).

Presentation of the gene symbols in Supplementary Table 1 derived from the heat map in Fig 4C is not really useful. It should at least indicate their cluster number.

Reviewer #2:

Remarks to the Author:

The data and text modifications made by the authors have improved the manuscript but several of the reviewer comments have not been adequately addressed.

1. As noted by reviewer #2 (Comment #7) and echoed by reviewer #1 (Comment #2), the possibility that the surviving cells are a type of progenitor cell fundamentally undercuts that premise of this manuscript. The authors provided new data (Suppl Fig 5) and softened their language in the discussion, but neither of these additions adequately address the reviewer concerns. The experiments included in Suppl Fig 5 were too short to support the authors claim of "remarkable stability in the frequency of survival ciliated cells" (Line 282) and the possibility remains that by 10-14 dpi surviving progenitor cells are already differentiating as part of the normal epithelial repair process.

2. The concern about CD24 as a marker for ciliated cells (reviewer #2, comment #4) has not been alleviated by the additional data added by the authors. The reviewer requested protein data and the authors provided immunostaining for Foxj1 but no protein data specifically for CD24. Also, while not definitive, multiple studies have identified CD24 as a cancer stem cell marker, so the possibility exists that CD24 may be a marker for progenitor cells. The use of a ciliated cell marker that is not widely accepted or supported limits the interpretation of the data in this manuscript.

Dumm et al., response to reviewers:

We thank reviewers for their review of the manuscript and additional comments. We have added new data and clarified several points in the text for increased clarity. Detailed responses to each specific point can be found below.

Reviewers' comments:

Reviewer #1 (Remarks to the Author):

1. For me, the demonstration is still not convincing. The observation of an expression of surfactant proteins does not fit with a ciliated cell origin. The association between CD24 and ciliated cell is not at all justified: to the best of my knowledge, it is usually said that CD25 is expressed in B lymphocytes, differentiating neuroblasts, neutrophils and their precursors (see Genecards, for instance), and I did not find a clear expression in ciliated cells. So the selection of cells based on this marker makes the demonstration very confusing to me. I understand that altered expression after infection can affect the expression of classical markers of ciliated cells, but a better choice of the bait seems to me mandatory and weakening the demonstration. I suspect that a better exploration of single cell RNA sequencing data would help clarifying the situation, even though I don't understand the use of the 2DPI condition. More information would have come from the latest time point, showing the survivors. With that in hand, it would be possible to use a program such as Monocle to build the lineage of these cells. That would certainly clarify a lot the demonstration. Unfortunately, at this point, the information provided by the single cell is not sufficient.

We apologize for not clearly defining our rationale for the use of CD24. In this revision, we have generated additional data (supplementary figures 3 and 4) validating CD24 (high) as a surface expressed flow cytometry marker for ciliated cells from an EpCam+ population. Specifically:

- We genetically labeled ciliated cells (FoxJ1-CreER knock-in line crossed to a tdTomato Cre-reporter line) and show that >95% are CD24+ (Sup Fig 3B-E).
- We performed protein staining on murine tracheal sections co-staining acetylated tubulin (apical cilia) and CD24 and show not only their labeling of the same cells, but also that CD24 labeling is specific for ciliated cells (Sup Fig 3F-G).
- We also performed protein staining on murine tracheal sections co-staining FoxJ1 and CD24 and show the same specific co-localization as observed with the apical cilia (Sup Fig 3H-I).
- Additionally, we show that CD24 continues to be expressed on FoxJ1/tdTomato positive survivor ciliated cells (Sup Fig 4G-H).
 - This is protein validation of our single-cell RNAseq experiment, which was included to show that CD24 is not lost in infected ciliated cells.
- Finally, we also cited 9 previous publications where CD24 is used as a flow cytometry marker for ciliated cells (line 203). A number of those papers: (*Balasooriya, G. I., et al. (2016), Dev Cell 37(1): 85-97. Wasserman, G. A., et al.*

(2017)," *J Clin Invest* 127(10): 3866-3876. Zhao, R., et al. (2014), *Dev Cell* 30(2): 151-165.) also included data validating CD24 as a ciliated cell marker in their supplemental material.

2. I agree with reviewer #1 that the diphtheria toxin experiment does not provide very clear-cut results.

We agree that this experiment has limitations. We have modified our discussion to be clearer that this experiment only shows that the elimination of survivor cells has a negative effect on virally induced disease (Lines 446-471).

Minor:

3. SftpA in Figure 4g is misspelled.

This has been corrected in the figure.

4. Moreover, page 14 probably refers to Supplementary Fig. 8C (Fig. 1C that does not exist!).

This has been corrected in the text to refer to Supplementary Fig 2C.

5. Presentation of the gene symbols in Supplementary Table 1 derived from the heat map in Fig 4C is not really useful. It should at least indicate their cluster number.

The origin of gene symbols in Supplementary Table 1 has been clarified in the figure legend. These genes do not have corresponding cluster numbers, as these genes resulted from the bulk RNA sequencing analysis, not the single-cell sequencing analysis.

Reviewer #2 (Remarks to the Author):

The data and text modifications made by the authors have improved the manuscript but several of the reviewer comments have not been adequately addressed.

6. As noted by reviewer #2 (Comment #7) and echoed by reviewer #1 (Comment #2), the possibility that the surviving cells are a type of progenitor cell fundamentally undercuts that premise of this manuscript. The authors provided new data (Suppl Fig 5) and softened their language in the discussion, but neither of these additions adequately address the reviewer concerns. The experiments included in Suppl Fig 5 were too short to support the authors claim of "remarkable stability in the frequency of survival ciliated cells" (Line 282) and the possibility remains that by 10-14 dpi surviving progenitor cells are already differentiating as part of the normal epithelial repair process.

We agree that we have no definitive experiments to show how this population of cells is being formed. This paper is focused primarily on defining the phenotypes of

these cells. We have clarified our discussion to point out the limitations of our experiments and acknowledge other potential models for their formation. (Lines 427-445).

7. The concern about CD24 as a marker for ciliated cells (reviewer #2, comment #4) has not been alleviated by the additional data added by the authors. The reviewer requested protein data and the authors provided immunostaining for Foxj1 but no protein data specifically for CD24. Also, while not definitive, multiple studies have identified CD24 as a cancer stem cell marker, so the possibility exists that CD24 may be a marker for progenitor cells. The use of a ciliated cell marker that is not widely accepted or supported limits the interpretation of the data in this manuscript.

Please see the description of our additional data and our detailed response under reviewer 1, comment 1.

Reviewers' Comments:

Reviewer #2:

Remarks to the Author:

The data and text modifications made by the authors have improved the manuscript but several of the reviewer comments have not been adequately addressed.

1. In response to Comment 2 of Reviewer 1, the authors softened the language of their interpretation of the results of the diphtheria toxin experiments (Lines 453-455) but the abstract still contains the more definitive, clear-cut language (Lines 67-69). The authors should modify the abstract to match the new additions to the discussion.

2. The authors provided additional data and citations to support the use of CD24 as a marker for ciliated cells and do demonstrate that ciliated cells express CD24 but the specificity of this marker within the lung is still unaddressed and is likely found in other nonciliated epithelial cell types. As noted by both reviewers, the use of a nonspecific ciliated cell marker limits the interpretation of the data in this manuscript.

3. For Supplemental Figure 4, the authors added staining data (panels G-H) on tissue (14dpi) to validate the single cell sequencing (2dpi) and in the author's words "to show that CD24 is not lost in infected cells". The authors should have included staining data for both 2dpi and 14dpi. Additionally, the suggestion by Reviewer 1 (Comment 1) of the inclusion of single cell sequencing data from 14dpi was not addressed in this revision and authors failed to even comment on the suggestion in the rebuttal.

4. In response to Comment 6 (Reviewer 2), the authors state the following:

We agree that we have no definitive experiments to show how this population of cells is being formed. This paper is focused primarily on defining the phenotypes of these cells.

The authors are not just "defining the phenotypes of these cells" but making very specific claims that these cells are "survivor" ciliated cells that fundamentally change their transcriptional output and become a distinct type of epithelial cell (Lines 62-65). Based on this response, this reviewer does not agree that the authors addressed Comment 6 with just some softening of the language in the discussion. If the authors are only interested in defining the phenotype of the "survivor cells", they should consider a much broader rewrite of the manuscript.

Dumm et al., response to reviewers:

We thank reviewers for their review of the manuscript and additional comments. Detailed responses to each specific point can be found below.

Reviewers' comments:

Reviewer #2 (Remarks to the Author):

The data and text modifications made by the authors have improved the manuscript but several of the reviewer comments have not been adequately addressed.

1. In response to Comment 2 of Reviewer 1, the authors softened the language of their interpretation of the results of the diphtheria toxin experiments (Lines 453-455) but the abstract still contains the more definitive, clear-cut language (Lines 67-69). The authors should modify the abstract to match the new additions to the discussion.

The abstract has been updated to match the additions to the discussion.

2. The authors provided additional data and citations to support the use of CD24 as a marker for ciliated cells and do demonstrate that ciliated cells express CD24 but the specificity of this marker within the lung is still unaddressed and is likely found in other nonciliated epithelial cell types. As noted by both reviewers, the use of a nonspecific ciliated cell marker limits the interpretation of the data in this manuscript.

To determine the specificity of CD24 as a marker for ciliated cells in the lung, we performed protein staining on murine tracheal tissue samples (Supplementary Figure 3). By staining CD24 and the canonical markers of ciliated cells (FOXP1 and Acetylated Tubulin), we demonstrated that CD24 co-staining is specific for ciliated cells in the upper airways and is not present on surrounding cell types that are not Foxj1+ and Acetylated tubulin+.

3. For Supplemental Figure 4, the authors added staining data (panels G-H) on tissue (14dpi) to validate the single cell sequencing (2dpi) and in the author's words "to show that CD24 is not lost in infected cells". The authors should have included staining data for both 2dpi and 14dpi. Additionally, the suggestion by Reviewer 1 (Comment 1) of the inclusion of single cell sequencing data from 14dpi was not addressed in this revision and authors failed to even comment on the suggestion in the rebuttal.

The reviewer is correct, our 2DPI analysis relied on transcriptional data and our 14DPI analysis relied on protein staining data. We have more precisely detailed which analysis was used at each timepoint for validating the use of CD24 in the main text.

4. In response to Comment 6 (Reviewer 2), the authors state the following:

We agree that we have no definitive experiments to show how this population of cells is being formed. This paper is focused primarily on defining the phenotypes of these cells.

The authors are not just “defining the phenotypes of these cells” but making very specific claims that these cells are “survivor” ciliated cells that fundamentally change their transcriptional output and become a distinct type of epithelial cell (Lines 62-65). Based on this response, this reviewer does not agree that the authors addressed Comment 6 with just some softening of the language in the discussion. If the authors are only interested in defining the phenotype of the “survivor cells”, they should consider a much broader rewrite of the manuscript.

We define these cells as “survivor” ciliated cells based on the presence (albeit diminished expression) of Foxj1, a canonical marker for ciliated cells that is well-documented. For clarity however, we have revised the manuscript to describe these cells as either “CD24+”, “FoxJ1+”, or “ciliated-like” survivor cells.